

# An Integrated Marine Data Collection for the German Bight – Part II: Tides, Salinity and Waves (1996 – 2015 CE)

Robert Hagen[1], Andreas Plüß[1], Romina Ihde[1], Janina Freund[1], Norman Dreier[2], Edgar Nehlsen[2], Nico Schrage[3], Peter Fröhle[2], Frank Kösters[1]

[1]Federal Waterways Engineering and Research Institute, Hamburg, 22559, Germany
[2]Hamburg University of Technology, Hamburg, 21073, Germany
[3]Bjoernsen Consulting Engineers, Koblenz, 56070, Germany

*Correspondence to* Robert Hagen (robert.hagen@baw.de, ORCID: 0000-0002-8446-2004)

## Abstract

The German Bight within the central North Sea is of vital importance to many industrial nations in the European Union (EU), which have obligated themselves to ensure the development of green energy facilities and technology, while improving natural habitats and still being economically competitive. These ambitious goals require a tremendous amount of careful planning and considerations, which depends heavily on data availability. For this reason, we established in close cooperation with stakeholders an open-access integrated, marine data collection from 1996 to 2015 for bathymetry, surface sediments, tidal dynamics, salinity, and waves in the German Bight for science, economy, and governmental interest. This second part of a two-part publication presents data products from numerical hindcast simulations for sea surface elevation, current velocity, bottom shear stress, salinity, wave parameters and wave spectra. As an important improvement to existing data collections our model represents the variability of the bathymetry by using annually updated model topographies. Moreover, we provide model results at a high temporal and spatial resolution (Hagen et al., 2020b), i.e. model results are gridded to 1,000 m at 20-minute intervals (10.48437/02.2020.K2.7000.0004). Tidal characteristic values (Hagen et al., 2020a), such as tidal range or ebb current velocity, are computed based on the numerical modeling results (10.48437/02.2020.K2.7000.0003). Therefore, this integrated, marine data collection enables coastal stakeholders and scientists to easily enter and participate in countless applications, which could be the development of detailed coastal models, handling of complex natural habitat problems, design of coastal structures, or trend exploration into the future.

## 1 Introduction

The North Sea on the northwest European Shelf is a contested region comprising interests of economic growth and the protection of future ecosystem services. On the one hand, ongoing developments of the European Union's blue growth initiative are e.g. a strong increase of the amount of energy produced by offshore wind farms and on the other hand, there is a strict legislation to ensure a good environmental status (e.g. by the Marine Strategy Framework Directive), which is monitored





closely. Pursuing both goals needs a reliable data base of hydrographical parameters to assess either economic prospects or ecological change.

The hydrography of the North Sea is characterized by the interaction of tides and surge from the North Atlantic, local wind and wave effects, and the interaction with adjacent estuaries (Otto et al., 1990). This short-term variability is overlain by long-
term changes such as sea level rise (Idier et al., 2017), spatially varying increase and decrease in tidal range (Jänicke et al., 2020; Müller, 2011), seasonality (Müller et al., 2014), and changing seabed morphology (Benninghoff and Winter, 2019; Winter, 2011). For this reason, hydrodynamic parameters of the North Sea are monitored by one of the densest measurement networks worldwide. Long-term measurements, such as sea surface elevation or salinity are bound to gauge locations whereas spatially more extensive measurements, such as ADCP campaigns, often lack temporal coverage. Remote sensing attempts to
bridge this gap, but still miss a temporal resolution in the order of individual tides.

Earth sciences and oceanography apply numerical, process-based models to fill data gaps for a user-specified model domain. Hindcast model data products for the German Bight started with the CoastDat project (Weisse and Plüβ, 2006), which has been subsequently updated to CoastDat2 (Geyer, 2014; Groll and Weisse, 2017). These data sets contain sea surface elevation, current velocity, and wave climate products at a regular 1.6 km spatial resolution with hourly time intervals (waves 5.5 km
regular grid, 3-hour intervals). Similarly to CoastDat2, the wind and wave climate is also described in the REA-40 data set which demonstrates higher skill when compared to measurements, although it is covering a shorter time period (Reistad et al., 2011). Similar approaches came from coastal engineering projects, e.g. the AufMod data collection (Kösters et al., 2014) which provides annual tidal characteristic values (i.e. tidal range, tidal high water etc.) and annual bathymetries on a 50 m raster from 1982 to 2012. Other data products cover the north-western European Shelf region (e.g. https://marine.copernicus.eu/), or the
entire globe (e.g. FES products by Lyard et al. (2006)) and are therefore limited to coarse grid resolution near the coast (minimum 2.5 km regular grids, usually much coarser, on the European Shelf).

As pointed out by Groll and Weisse (2017) and Rasquin et al. (2020), the spatial resolution of data sets in the German Bight must be high in order to properly resolve the morphologically complex nearshore area in the German Bight, which contains islands, extensive tidal flats and deep channels with a typical width of 10 km to less than 1 km. None of the data products
mentioned above reach that resolution. Furthermore, a typical tidal cycle in the North Sea takes about 12.4 hours and contains two peaks in current velocity (flood and ebb). Hence, it can be argued that a 1-hour resolution is too coarse to represent peaks in both, sea surface elevation and current velocity. Additionally, almost none of the hindcasts above use annually varying bathymetry for numerical model simulations which is crucial in the morphodynamically highly active Wadden Sea (Benninghoff and Winter, 2019; Winter, 2011), as bathymetry variation in the nearshore area has shown to have an impact on
large-scale tidal dynamics (Jacob et al., 2016).

For further progress in terms of spatial and temporal resolution / coverage, a 20-year hindcast marine data collection for the time from 1996 to 2015 based on numerical modeling results with annually-updated bathymetry is established here as the second part of a two-part publication. Included data products are sea surface elevation, current velocity, bed shear stress, salinity, wave parameters, and wave spectra in the German Bight at a high spatial (1,000 m by 1,000 m) and temporal (20-



minute intervals) resolution. The first part of this publication focusses on annual bathymetries and surface sediment data (Sievers et al. (2021), subm.) in the German Bight. Numerical modelling at this temporal and spatial scale has become possible due to the availability of high-resolution bathymetry, surface sediments, reanalyzed meteorology (Bollmeyer et al., 2015), and input from global modelling products such as FES. To provide an optimal usability of our data set stakeholders from scientific, commercial, and governmental communities has been involved, which led to additional data products (i.e. annual tidal

characteristic and harmonic analysis).

This paper focusses on the description and validation of the newly created data sets. In Sect. 2 the product lineage from numerical model set-up and output, analysis methods for the computation of data products is outlined. Selected data are shown for illustration in Sect. 3. In Sect. 4 numerical model data are validated against field measurements. A list of all data products is given in appendix 10.2.

**2   Methods**

**2.1   Product Lineage**

To ensure traceability (origin and subsequent processing history) and to avoid unintended misinterpretation of our marine data collection products, a documentation of data product lineage was carried out. This information is an essential part of our products meta data and provides the ability to retrieve and understand the relationship between input data, model- or analysis

results, and data products (Figure 1).

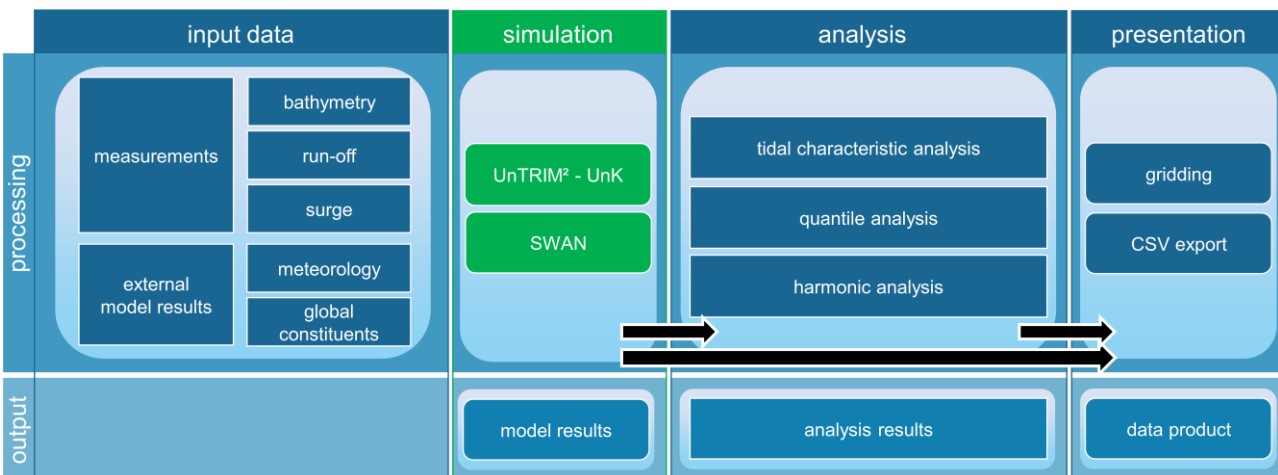

**Figure 1: Schematic overview over product generation and lineage covering the overall modelling steps from input through simulation, analysis and presentation.**

While unprocessed model results are in the range of terabytes (2.8 TB/a), analyses results are in a medium gigabyte range (100 GB/a) which makes them more manageable and user-friendly. For the final presentation, model and analysis results are gridded



to data products at varying spatial resolutions and extent (i.e. Figure 2) for further applications such as on-the-fly web viewer interaction and GIS operations. These gridded files represent the data products within this data collection and are in the low megabyte to gigabyte range (15 MB to 17 GB). Additionally, full wave spectra are extracted at representative locations via
CSV-export as data product. All products are distributed to users as offline (file-based) and online solutions i.e. web map service (WMS), web feature service (WFS), or on-the-fly visualization via THREDDS data server). We supply additional wave spectra information at chosen locations on the EasyGSH-DB product zone (EPZ) and the 20 mNHN (NHN: German Chart datum) isobath which can be used for nesting applications in numerical modeling. Our spatial data products are distributed on regular grids (in the 12-SM, EPZ and EEZ from Figure 2) which are in the German exclusive economic zone
(EEZ), the EasyGSH-DB product zone (EPZ) or the 12 nautical miles zone of the German authorities (12-SM). Gridded model results include sea surface elevation, depth-averaged current velocity (northward, eastward), significant wave height, mean, peak, TM1 and TM2 wave periods, mean wave direction, wave directional spreading, depth-averaged salinity, and bottom shear stress (northward, eastward) in 20-minute intervals for the period from January, 1$^{st}$ 1996 to December, 31$^{st}$ 2015. Tidal characteristic values (i.e. tidal range etc.) and harmonic analysis results are provided as separate annual files. Furthermore,
every data product is documented via inspire-conform meta data, including the product specific lineage.

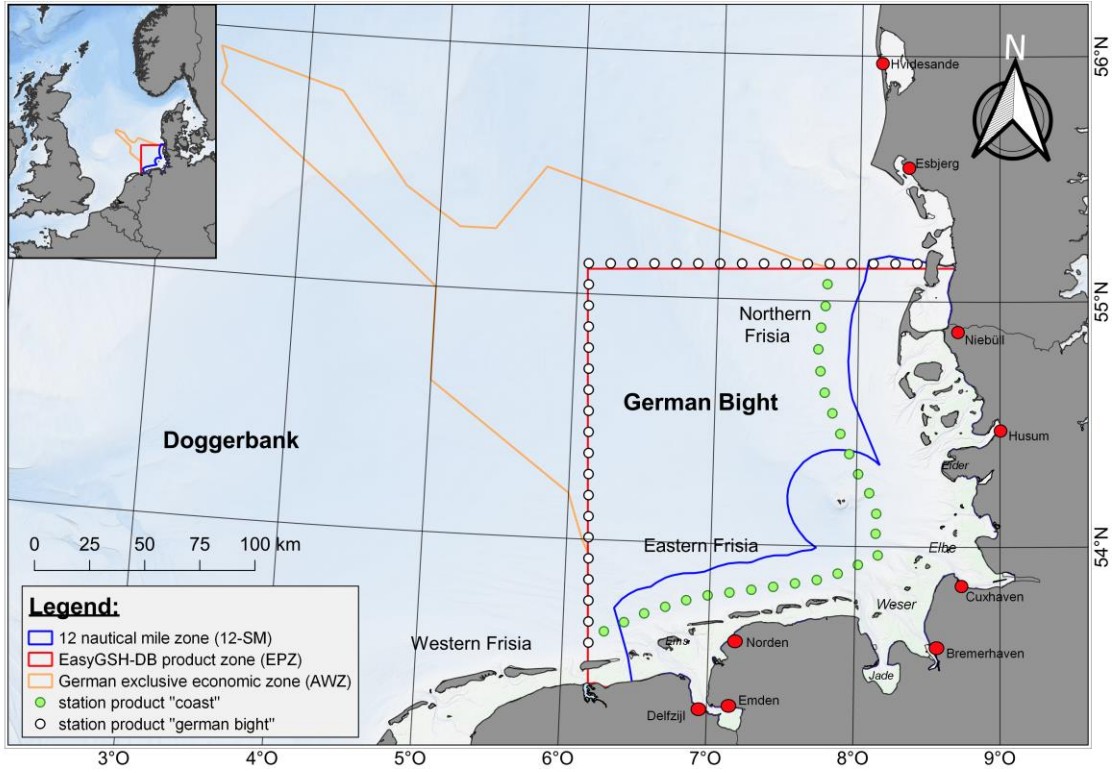

**Figure 2:** Outline of product polygons for data products in the Southern North Sea. The German 12 nautical miles zone (12 SM) is shown in blue, the EasyGSH-DB product zone (EPZ) in red, and the German exclusive economic zone (EEZ) in orange. Green and
white dots represent wave spectra results in the coast (CZ) and the EasyGSH-DB product zone (EPZ).



A horizontal grid-spacing resolution had to be chosen which (a) minimizes the raster error of the original data, (b) is still performant in a web environment, (c) is manageable in offline applications, and (d) uses an acceptable amount of disk space speaking from data creator and the user's perspective. Thus, the varying spatial extent and grid-spacing was selected based on physical considerations, manageability, and stakeholder feedback. We chose a regular grid spacing of 1,000 m for all

simulation results in the EPZ, as a finer resolution would increase data size tremendously. Annual data products (e.g. analyses products) use a higher spatial resolution of 100 m in the EPZ and the 12-SM zone and of 1,000 m in the EEZ zone. These products are provided separately in a common GeoTIFF format which can be processed by conventional GIS software. All simulation results were transformed into a state-of-the-art, structured NetCDF format and separated into annual sea surface elevation, current velocity, salinity, bottom shear stress, and wave data to manage file size. The time interval of these products

is reduced to 20-minute intervals due to limitations regarding commonly available web visualization software which does not allow more than roughly 32,000 time steps (restrictions on max integer length).

## 2.2 Numerical Modelling

The wide range of products and data analyses require a computational model setup to be (a) consistently applicable for all 20 years, (b) sufficiently detailed concerning horizontal and vertical mesh resolution (c) representing all necessary physical

processes, and (d) computationally efficient. For this reason, boundary and initial data sets for water level, bathymetry, wind speed, air pressure, and fresh water discharge must be available for the entire modeling time span from 1996 to 2015 to keep data products consistent.

We apply the modelling systems UnTRIM² with the novel subgrid approach for high-resolution bathymetry representation on unstructured grids (Casulli, 1990; Casulli and Stelling, 2011) for the simulation of tidal dynamics and transport, and the

sediment transport module SediMorph (Malcherek et al., 2002) for bottom roughness estimation. Waves are computed by the unstructured k-model UnK (Schneggenburger et al., 2000) and SWAN. The modelling approach considers 3D hydrodynamics, waves, daily freshwater discharge, hourly wind forcing and air pressure fluctuation, external surge from the Northern Atlantic, and the transport of salinity and heat flux. The open boundaries to the North Atlantic are forced with tidal constituents from FES 2014b (FES 2014 was produced by Noveltis, Legos and CLS and distributed by Aviso+, with support from Cnes

(https://www.aviso.altimetry.fr/)) for astronomical water levels, constant salinity and a characteristic, monthly temperature averaged over the entire water column. The initial water levels at the open boundaries from tidal constituents are then corrected for surge based on water level differences between calibrated simulations and nearby observations (Plüß, 2003). This approach implies that surge is constant along an open boundary which is not the case in nature. However, modeling practice has shown that sea surface elevation agrees well in the German Bight despite this simplification. Further aspects concerning the numerical

model, the calibration procedure and a thorough validation are published separately in BAW Technische Berichte et al. (2020). The model domain (Figure 3) covers the North Sea from Norway to Scotland, the English Channel, and the Danish Straits. Major estuaries in the German Bight, such as Ems, Weser, and Elbe, are included up to their tidal weirs. The model extends approximately 1,400 km in north-south and 1,200 km in west-east direction, respectively. Previous modeling approaches





(Kösters et al., 2014; Plüß, 2003; Putzar and Malcherek, 2015) have shown that tidal dynamics and transport in the German
Bight can be reproduced well when using the entire North Sea or the entire European continental shelf (Zijl et al., 2013) as
these large-scale approaches explicitly resolve tide-surge interaction and the composite amphidromic system of the North Sea.
The model uses an unstructured grid with a varying horizontal grid resolution of 10 km near the northern boundary down to
45 m in the Ems estuary, with roughly 75 % of all grid nodes located in the German Bight. The German Wadden Sea and the
outer estuaries of Ems, Weser and Elbe are resolved with a typical edge length between 180 m and 500 m (see Figure 3 for an
example of the resolution in the Wadden Sea). Additionally, a subgrid refinement is applied which additionally increases
volume approximation of the grid substantially at relatively low computational cost (Casulli, 2009; Sehili et al., 2014). The
subgrid refinement ensures that the constantly varying annual bathymetry is represented in the model. Here we applied the
subgrid approach with a refinement factor of 4 (open North Sea) up to 12 (within the estuaries). This discretization results in
roughly 202,000 horizontal grid- and 10,000,000 subgrid elements. The vertical discretization utilizes 54 fixed z-layers with a
half meter resolution between +4 and -20 mNHN, gradually becoming coarser downwards.

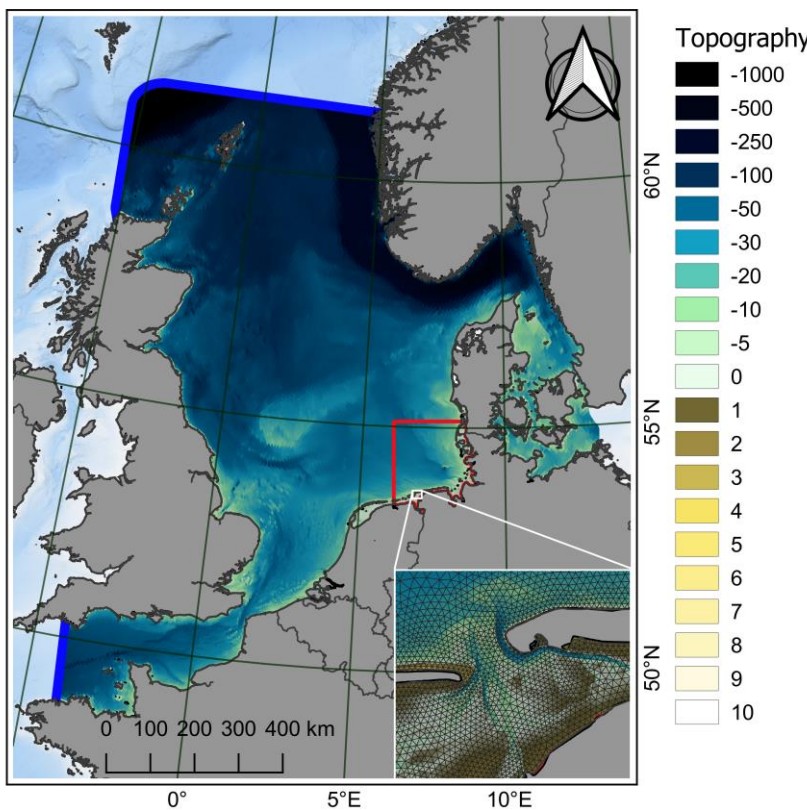

**Figure 3: The grid of the numerical North Sea model showing the grid in grey, the open boundaries in blue, the EasyGSH-DB product zone (EPZ) for scale in red, and a zoom to a part of the Wadden Sea (Eastern Frisia with Juist and Norderney Island).**
**Topography is given with negative values indicating depths with respect to German Chart datum (mNHN).**

Wind speed and air pressure fluctuation are extracted from the COSMO-REA6 data set (Bollmeyer et al., 2015) which provides hourly, reanalyzed meteorological data at a 6 km resolution. Fresh water discharge has been considered at the Dutch coast (Rhine, Maas, Ijsselmeer, and Waal river), and for the major estuaries in the German Bight (Ems, Elbe, Weser, and Eider) together with their main confluents run-off (Leda, Wümme, Lesum, Hunte). Dutch freshwater discharge was obtained from

https://waterinfo.rws.nl/ and run-off data for the German freshwater discharge was requested from the responsible authorities BfG, WSV and BSH. Annual bathymetries from Sievers et al. (2021, subm., e.g. Figure 4) are interpolated on the computational subgrid annually within the EPZ (i.e. Figure 2) and parts of the Dutch Wadden Sea.

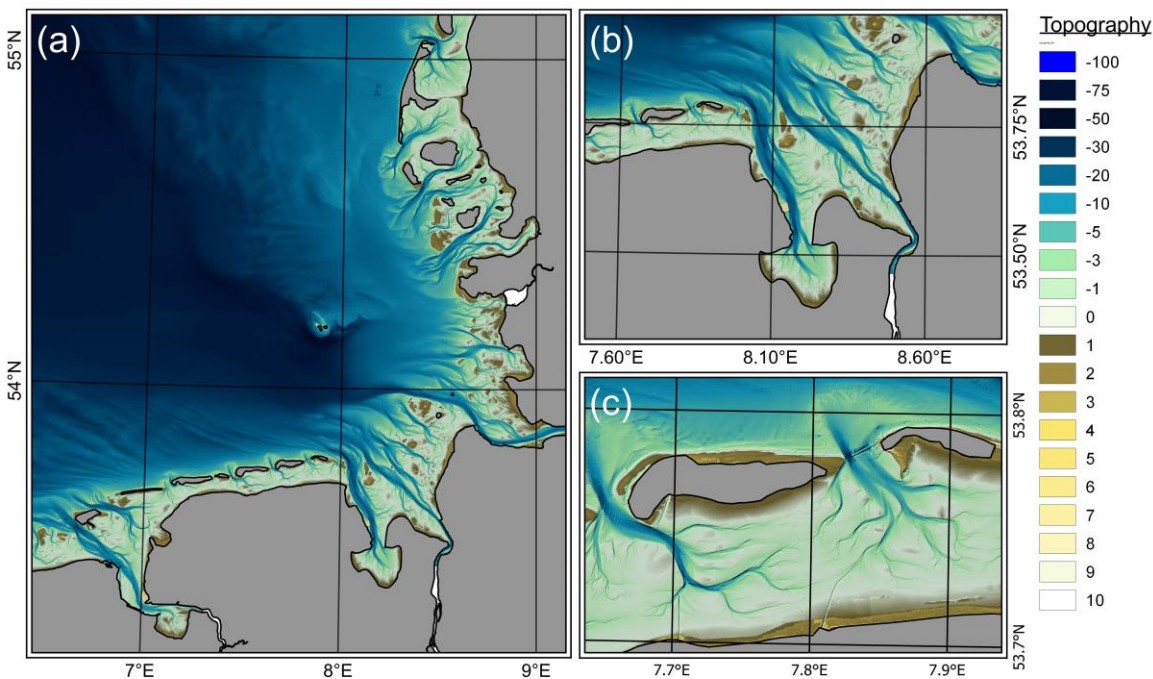

**Figure 4: Topography in 2015 at the native 10 m resolution in the German Bight (a), the Jade and Weser estuary (b), and the**
**Spiekeroog- and Wangerooge inlets (c). Topography is given with negative values indicating depths with respect to German Chart datum (mNHN).**

The remaining bathymetry of the North Sea was obtained from Rijkswaterstaat (https://inspire.caris.nl/viewer/), UKHO (https://datahub.admiralty.co.uk/), SHOM (https://data.shom.fr/), and EMODnet (EMODnet Bathymetry Consortium, 2018). The bathymetry outside the EPZ was assumed to be constant over time. Data from external sources were checked semi-
automatically and if necessary corrected for outliers, errors in unit or vertical coordinate reference system before usage.

Major groins, dams and training walls are included in the model grid at their realistic height and extent in the German Bight. Bottom roughness has been calibrated using spatially varying Nikuradse roughness ranging from 0.08 m in the English Channel to 0.002 m in Northern Frisia. Turbulence closure uses a conventional k-ε model with constant values for horizontal and vertical viscosity. Initial conditions for water level, current velocity, waves and the transport of salinity and heat are nested





from model results of predecessor years. The first year (1996) was started from an astronomically forced simulation using FES2014b without a surge correction of 1995 which was initialized using the initial salinity and temperature distribution from a climatology of Janssen et al. (1999).

Waves are computed using the models UnK and SWAN. The UnK wave model is run on a separate, unstructured grid which locates more (roughly 80 %) of its elements between the -30 mNHN isobath and the coastline of the German Bight. The

horizontal grid resolution for waves varies between 20 km near the open boundary to 150 m in the tidal channels of the German Wadden Sea. The wave spectrum is limited to 32 frequencies between 0.006 and 1.6 Hz with 24 specified directions (steps of 15 °). UnTRIM² and UnK are two-way-coupled which implies that current velocity, water level and meteorological forcing are communicated between the models at every time step.

Additionally, nonstationary wave simulations with SWAN are carried out on an unstructured grid which consists of 580,000

elements with 300,000 nodes in total, and applies the same boundary data as the UnTRIM²-UnK modeling approach at mean water level. The model domain includes the area from Figure 3 with a similar resolution between 50 m at the coast to 400 m near the -20 mNHN isobath in the German Bight up to 2500 m in the open North Sea. Due to the absence of water level and current interaction and therefore much lower computational cost, a more detailed discretization of the wave spectrum could be chosen. Wave spectra are resolved at 144 directions (2.5 °) using 42 frequencies (0.02 to 1 Hz). The wave model accounts for

exponential wave growth due to wind exposure (Komen and Hasselmann, 1984), white capping (Komen and Hasselmann, 1984), depth-induced wave breaking (Battjes and Janssen, 1978), bottom friction (Hasselmann et al., 1973), nonlinear wave interactions in deep and intermediate water depth due to quadruplets (Hasselmann and Hasselmann, 1985), and triads (Eldeberky and Battjes, 1996).

### 2.3 Data Analyses

Coastal engineers often reduce complex information, such as sea surface elevation or salinity signals, to meaningful, characteristic parameters (e.g. tidal range) through harmonic or tidal characteristic analysis. Tidal characteristic values define the behavior of periodic data with mean and extreme values in a tidal context, while harmonic analysis derives amplitude and phase for predefined (tidal) frequencies from a water level or current signal. For tidal characteristic and harmonic analysis presented hereafter, the program NCANALYSE (https://wiki.baw.de/en/index.php/NCANALYSE) is applied from January to

December for each modelled year. Note that the nodal f-u correction of tidal constituents has been disabled in the harmonic analysis due limited applicability in the German Bight (Hagen et al., 2021).

Our tidal characteristic analysis extends a classical Eulerian analysis approach by interpreting the entire model domain altogether in a Lagrange-like way (Figure 5). This is advantageous, because this approach guarantees that every tide and tidal parameter in the domain is related to the same event (e.g. a tidal cycle). This procedure yields consistent characteristic values

even for large domains, as each tide is linked to its predecessor. Hence, the transition from local to spatial characteristic values becomes feasible. The analysis starts by identifying each tide (i.e. times of high water and low water) within a given analysis time span for a main reference position (black dot, Figure 5). Starting from there, a phase difference (M2 constituent only)

between two adjacent locations for a chain (directed graph) of additional reference positions (red dots, Figure 5) is determined. M2 phase differences are finally converted to approximate travel time of the tidal wave between neighboring positions. This

procedure enables us to follow the same event (i.e. tidal cycle) throughout the domain by means of shifting the data analysis period, originally given for the main position. Finally, the data analysis period of the nearest reference location is used to determine e.g. high water, low water, time of high and low water, mean water level, etc. Lagrange-like tidal characteristic analysis can be performed for sea surface elevation, current velocity, salinity, and bed shear stress by linking these parameters to the tide. In addition, quantiles can also be calculated from the tidal characteristic values of the individual tides. In addition

to this, a harmonic analysis for the dominant semidiurnal moon tide M2 from the sea surface elevation as well as quantile analysis were carried out for water level and salinity.

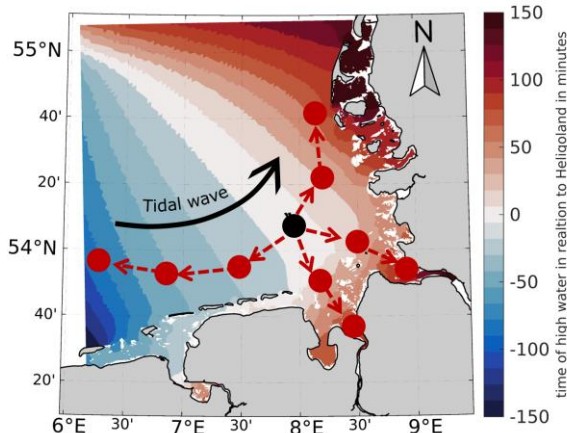

**Figure 5: Schematic overview of the tidal analysis' methodology throughout the German Bight, showing the main reference position in black and the predecessor and successor positions in red.**

Wave data analysis is performed via basic data operations, such as annual quantile or averaging, and has been carried out spatially for SWAN and UnK results.

## 3    Data Description

### 3.1    Tidal Dynamics and Salt Transport

Figure 6 shows an exemplary, chosen hydrodynamic state with sea surface elevation in mNHN (a), north- and eastward current

velocity in m/s (b), salinity in ppt (c), and north- and eastward bed shear stress in N/m² (d) on November 17th, 2015 7:00 PM. We can see the low tide approaching from the west in Eastern Frisia and its eastward propagation towards the mouth of the Elbe estuary (a), which results in northwestwards currents at this phase of the tide (b), with current velocity above 1.0 m/s in the tidal channels and estuaries. The outer German Bight shows a salinity (c) between 30 and 35 ppt, while the estuaries range between 3 and 25 ppt. Bed shear stress (d) shows seaward directed values near the Elbe, Weser, and Ems estuary. Bed shear

stress is shown for the 12-SM zone only, values in the deeper parts of the German Bight are almost negligible due to low
magnitude.

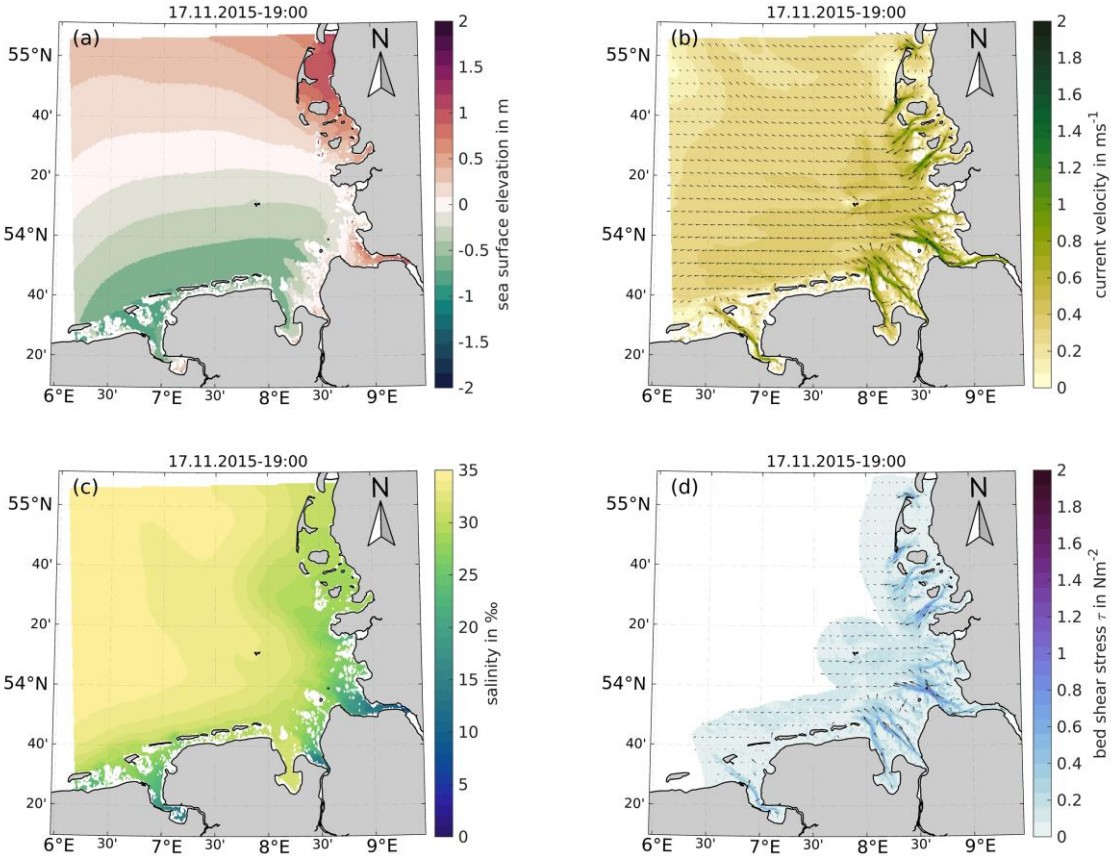

**Figure 6: Exemplary hydrodynamic state on 17. November, 2015 7:00 PM, showing sea surface elevation (a) in mNHN, current velocity magnitude and direction (b) in m/s, salinity (c) in ppt and bottom shear stress magnitude and direction (d) in N/m².**

The components of the hydrodynamic state shown in Figure 6 can be extracted at any point and time between January 1996
and December 2015 at the spatial and temporal resolution of the EasyGSH-DB data set of 1,000 m and 20 minutes, respectively.
Note, that gridding of unstructured model results, decreases data accuracy behind the islands of the Wadden Sea and in outer
estuaries, because 1 km is coarser than many of the narrow channels inside the Wadden Sea, resulting in a misrepresentation
of wetting and drying. For this reason, the annual inundation period is supplied as an analysis product at a 100 m resolution,
so that points affected by reoccurring tidal wetting and drying may be quickly identified by users. An overview of all synoptic
model simulation products is provided in the Appendix as Table A1.

### 3.2  Waves

Gridded UnK wave products are shown exemplarily for the significant wave height $H_{m0}$, mean wave direction $\Theta_m$ and peak
period $T_p$ in Figure 7 during a storm "Xaver" in December, 2013. Significant wave heights above 5 m are present in the deeper

parts of the German Bight which decline quickly, when reaching the nearshore areas and the Wadden Sea. UnK wave products include the significant wave height $H_{m0}$ in meters (filled contours in Figure 7, a), the mean wave period $T_{m02}$, the peak period $T_p$ both in seconds, the mean wave direction $\Theta_m$ (vectors in Figure 7, a) and the directional spread $\Psi_m$ in degree in 20-minute intervals.

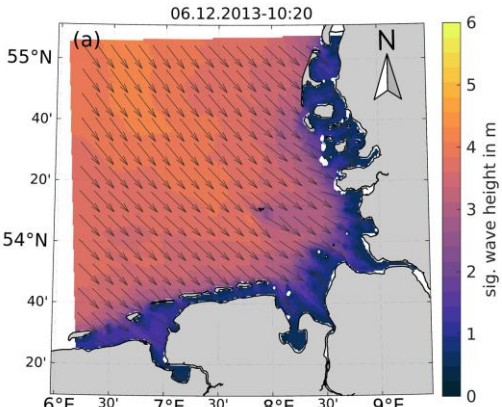 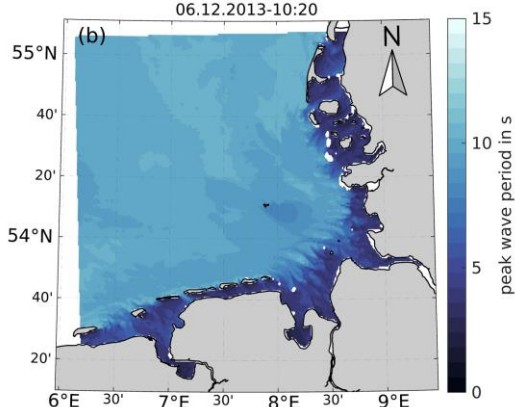

**Figure 7: Exemplary wave state during a storm "Xaver", in 2013 showing significant wave height $h_{m0}$ in meters, the mean wave direction $\Theta_m$ (a) and peak wave $T_p$ period in seconds (b) in the EPZ. Note that the mean wave direction vectors in (a) are normalized.**

SWAN wave spectra are compiled at the outer boundary of the EPZ at selected locations (shown in Figure 2, white dots). Hourly directional energy density wave spectra and time series of the wave parameters $H_{m0}$, mean and energy wave period ($T_{m02}$ and $T_{m-1.0}$), peak period ($T_p$), mean wave direction ($\Theta_m$) and directional spread ($\Psi_m$) are provided at those locations.

SWAN wave spectra may be used in addition to the time series of wave parameters from SWAN or UnK, e.g. as forcing wave boundary data for numerical wave models (nesting approach) or trend analysis of wave parameters. An exemplarily directional wave energy density spectra during "Xaver" (at FINO1) is also provided in the supplementary material S6 to this paper.

### 3.3  Model Data Analyses

This section describes exemplary data analysis products for tidal dynamics, sea state, and salinity which were calculated based
on model results. Figure 8 visualizes analyses product examples of the 50% quantile of the tidal range (a), the 50% quantile of the ebb current velocity (b), the 50% quantile of the tidally averaged salinity (c), and the ratio of the mean flood to the mean tide current velocity (d) for the EPZ as annual averages of the year 2015. While the tidal range lies below 1 m at the north east end of the German Bight due to proximity to an amphidromic point, the tidal range increases towards the coast before reaching a maximum in the Jade Bay and within the estuaries. The mean ebb current velocity ranges between 1 and 1.5 m/s in the deeper
channels and between 0.25 to 1 m/s in the offshore areas of the German Bight. The annually, tidally averaged salinity reflects the influence of the fresh water supply of the adjacent estuaries in the German Bight, with a decrease in salinity in the mouth and downstream of the estuaries. The ratio of the mean flood to mean tide current velocity varies at small spatial scale near the coast but indicates a general flood dominance in Eastern Frisia which declines in the ebb delta shores of the barrier islands and



the main estuaries. Northern Frisia demonstrates different behavior with an overall balanced ratio of flood to tide current

velocity.

We decided to provide quantiles for scalar and mean / maxima for vector tidal characteristic values from the tidal analyses' products, to avoid a distortion due to e.g. the effect of the storm surges. Extreme values for sea surface elevation and salinity are given by the 1 % and 99 % quantile of the annual simulation results. In addition, we provide the number of tidal high and tidal low water events and the mean inundation period for every year. Current velocity and bed shear stress are processed

accordingly. The ratio between flood- and ebb- to tide velocity is calculated for the quantification of current asymmetry. The harmonic analysis includes the amplitude and phase of the semidiurnal moon tide M2 (without nodal modulation, see Sect. 2.3). An overview of all analysis products is provided in the Appendix as Table A3.

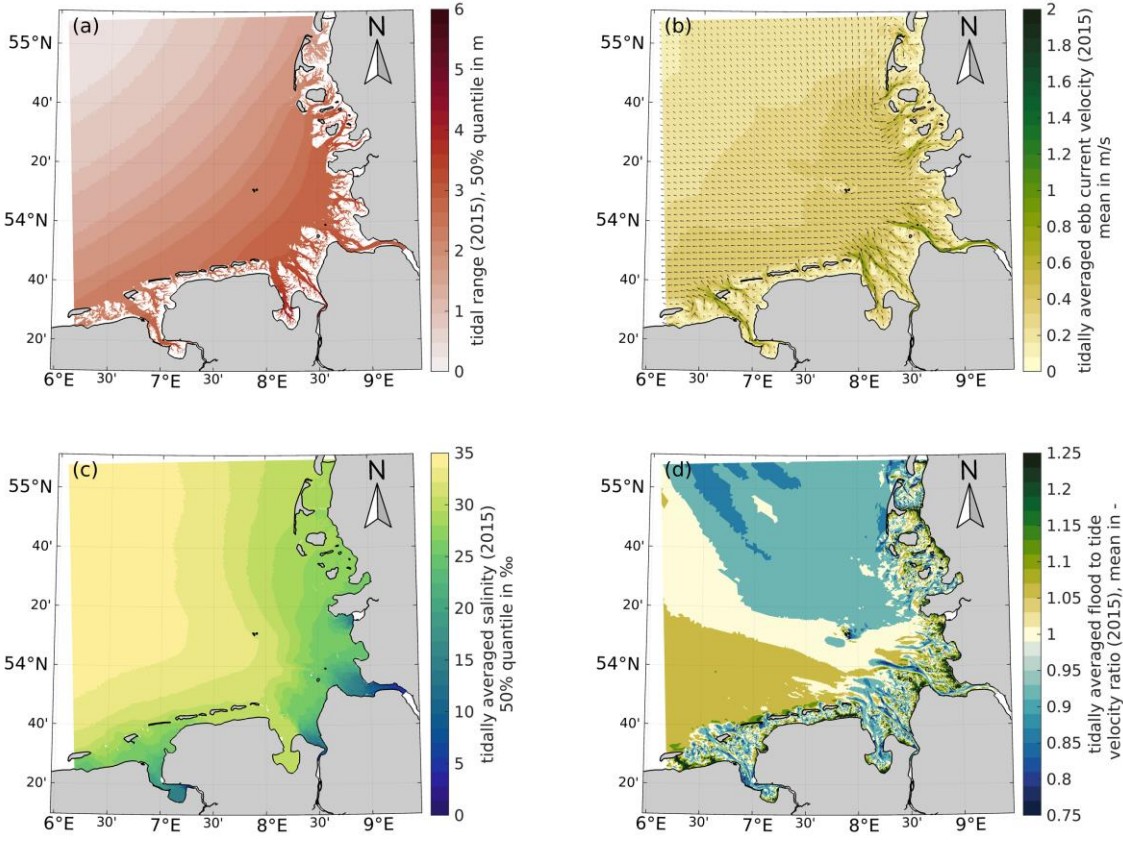

**Figure 8: Examples of tidal characteristic values for 2015: (a) 50% quantile of tidal range in m, (b) mean, tidally averaged ebb**
**current velocity in m/s, (c) mean salinity in ppt and (d) ratio of mean flood and mean tidal current velocity in -.**

Wave analysis products, such as quantiles and the maximum of the significant wave height $H_{m0}$ the mean wave period $T_{m02}$ at max. significant wave height etc. have been calculated annually from UnK and SWAN model results. SWAN products include the annual mean peak wave period, mean $H_{m0}$, mean wave energy density and a cumulative analysis of wave parameters at the "coast" and "German Bight" stations (see Figure 2), and the energy weighed mean wave direction (i.e. wave propagation



direction as defined in IAHR (1989)) for the EPZ. Further wave analysis products have been compiled based on the SWAN simulation results at selected locations near the -20 mNHN isobath (shown in Figure 2, green dots), e.g. for coastal protection applications. As an example, the annual combined frequency of occurrence for the significant wave height and the mean wave direction is shown for one selected location in the supplementary material S5.

## 4    Validation

In the following, we show the model validity for the years 1996 to 2015 using harmonic and tidal characteristic analysis for observed and modelled data. Waves, current, and salinity are validated against measurements in the EasyGSH-DB product zone (EPZ, see Figure 9) using error metrics provided in the Appendix. All measurements have been checked visually and if possible corrected for outliers and suspect data points. Applied measurement locations are provided in Figure 9. A full validation of the UnTRIM2 modelling approach is documented in BAW Technische Berichte et al. (2020) for the years 2006 and 2012. In addition, short, annual validation documents (e.g. BAW Technische Berichte et al. (2019)) are available for each year of our hindcast period of 1996 – 2015. Observational data were obtained from local authorities (see supplemental material S1), as the marine data collection described hereafter excludes observational hydrodynamic data.

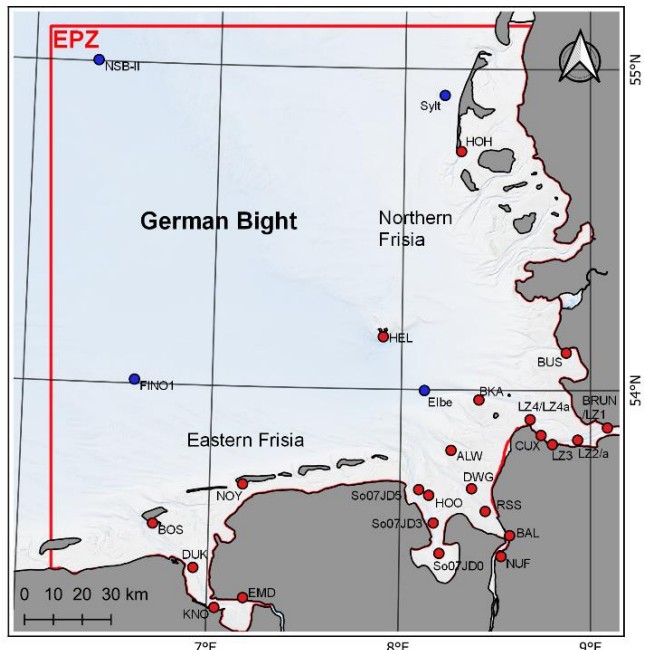

**Figure 9: Gauge map in the German Bight showing the gauge locations (red dots) and wave gauge locations (blue dots).**





## 4.1 Tides

Through harmonic analysis, a water level signal can be reduced to several harmonic components with varying amplitude, phase, and frequency. As tides originate from the gravitational forces of the sun, moon, and earth itself, the frequency of each driving force is clearly defined. A tidal constituent therefore represents the amplitude and phase lag of a predetermined

astronomic frequency (e.g. the semidiurnal moon tide M2). The sum of all constituents is referred to as astronomical tide and the methodology is described extensively in literature (Codiga, 2011; Pugh, 1987). In the following, the semidiurnal moon tide M2 was chosen, as its amplitude is more than 7 times larger than any other constituent in the German Bight, making it the dominant driving force of tides.

An annually varying network of 10 (1996) to 41 (2006) tide gauges in the model domain is used for validation with most

gauges being inside the EasyGSH product zone (EPZ). The varying number of gauges results from limited data availability, and quality restrictions of water level records. Measurements have been checked visually and if possible corrected for outliers and suspect data points. The measured and predicted water levels are analyzed harmonically (methodology see Sect. 2.3) and the differences (errors) between predicted and observed amplitude and phase are used for error metrics. We apply a mean error; a standard deviation, and a root mean square error (RMSE) for a goodness-of-fit estimation of M2 amplitude and phase

for each year in Table 1.

**Table 1: Mean error (ME), standard deviation (σ), and root mean square error (RMSE) of amplitude A in m and phase g in ° of the M2 tidal constituent for years 1996 to 2015. Note that the number of gauges (# gauges) available varies in individual years.**

| year | # gauges | A in m | | | g in ° | | |
|------|----------|--------|--------|------|--------|--------|------|
| | | ME | σ | RMSE | ME | σ | RMSE |
| 1996 | 10 | -0.005 | ±0.04 | 0.03 | -3.882 | ±2.46 | 4.53 |
| 1997 | 12 | -0.003 | ±0.04 | 0.04 | -2.189 | ±2.68 | 3.37 |
| 1998 | 12 | -0.027 | ±0.04 | 0.05 | -1.217 | ±1.23 | 1.69 |
| 1999 | 19 | -0.001 | ±0.05 | 0.05 | -2.499 | ±1.80 | 3.05 |
| 2000 | 25 | -0.018 | ±0.03 | 0.04 | -2.962 | ±1.27 | 3.21 |
| 2001 | 28 | -0.006 | ±0.04 | 0.04 | -2.384 | ±1.75 | 2.94 |
| 2002 | 27 | -0.023 | ±0.04 | 0.04 | -1.727 | ±1.71 | 2.40 |
| 2003 | 23 | 0.001 | ±0.04 | 0.03 | -1.461 | ±2.04 | 2.47 |
| 2004 | 27 | -0.022 | ±0.03 | 0.04 | -2.309 | ±1.91 | 2.97 |
| 2005 | 30 | -0.003 | ±0.04 | 0.04 | -4.881 | ±1.72 | 5.16 |
| 2006 | 41 | 0.012 | ±0.07 | 0.07 | -2.323 | ±3.28 | 3.99 |
| 2007 | 32 | -0.024 | ±0.05 | 0.05 | -1.494 | ±1.91 | 2.40 |
| 2008 | 28 | -0.030 | ±0.04 | 0.05 | -1.749 | ±2.46 | 2.98 |
| 2009 | 25 | 0.007 | ±0.04 | 0.04 | -2.187 | ±2.69 | 3.43 |
| 2010 | 30 | -0.002 | ±0.04 | 0.04 | -1.804 | ±2.15 | 2.78 |
| 2011 | 27 | 0.003 | ±0.04 | 0.04 | -3.113 | ±2.28 | 3.83 |
| 2012 | 26 | 0.003 | ±0.04 | 0.04 | -2.502 | ±2.22 | 3.31 |
| 2013 | 31 | 0.008 | ±0.03 | 0.03 | -2.886 | ±2.22 | 3.62 |
| 2014 | 23 | 0.015 | ±0.03 | 0.04 | -3.267 | ±1.72 | 3.67 |
| 2015 | 25 | 0.021 | ±0.04 | 0.04 | -3.335 | ±1.91 | 3.82 |





The mean error ranges between -3 and 2.1 cm with a standard deviation of 3 to 7 cm. The largest RMSE is calculated in 2006 with 7 cm and the lowest in 1996, 2004, 2013 and 2014 with approximately 3 cm. It must be noted that 2006 is the calibration 320 year. For this year additional gauges outside of the focus area were considered, as well. This led to worse error metrics as model resolution and calibration efforts are reduced outside the EPZ. The mean phase error is between -1.2 and -4.9° in 2005 and 1998, respectively, and the standard deviation ranges between 1.3 to 3.3°, indicating good agreement between observation and prediction. The RMSE of the phase does not exceed 5.2 ° (2005) which would correspond to a M2 phase lag of 10 minutes. Comparable North Sea modelling approaches from literature show M2 RMSEs between 6.4 to 20 cm for the amplitude and 325 5.1 to 10 ° for the phase (Gräwe et al., 2016; Jacob et al., 2016; Plüß, 2003; Zijl et al., 2013), showing that our validation results compare well to the benchmarks in literature.

As documented in Table 1 that the model reproduces astronomical tides, we can compare observed and modelled tidal signals through their tidal characteristic values throughout the data set. Again, the RMSE is applied for each year with an average of 705 tides per year. Figure 10 displays the RMSE distribution for the tidal range at chosen gauges (for location see Figure 9) 330 throughout the EPZ. If more than 50 tides per year are invalid, e.g. due to missing or inconsistent observational data, no RMSE is calculated. No-data values in Figure 11 may therefore be explained by a lack of observed data, large data gaps, a high number of suspicious values, or outliers in the measurements. The scale was chosen to a maximum of 5 % and a minimum of 1 % of a typical macrotidal range of 5 m in the German Bight.

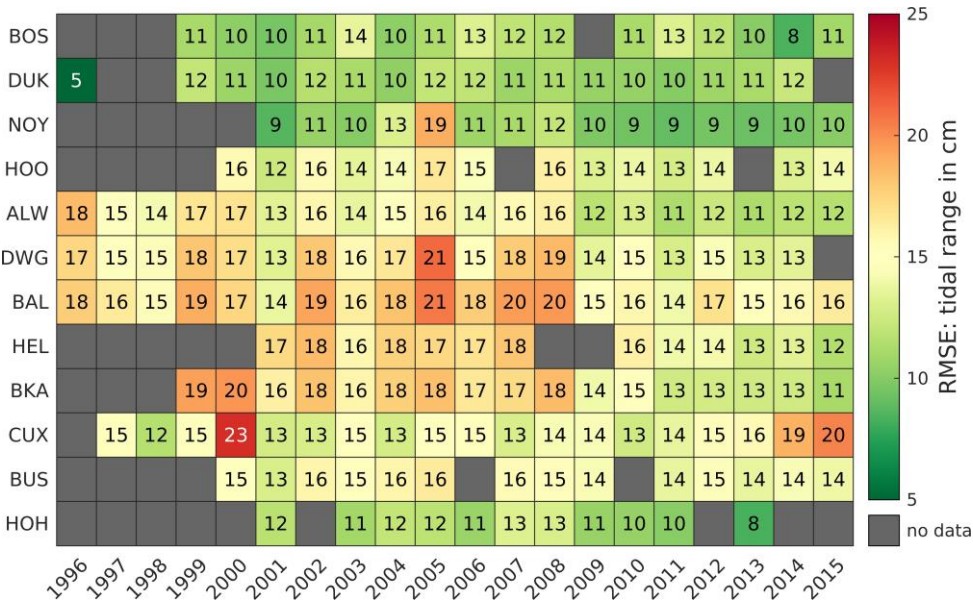


**Figure 10: Root mean square error (RMSE) of the tidal range in cm between 1996 to 2015 at representative gauges.**

Most RMSEs of tidal range are between 10 to 20 cm, except for DWG, CUX and BKA, usually before 2008. The RMSE is lowest at DUK, NOY and HOH with 5, 9 and 8 cm, respectively, and largest at DWG, HEL and CUX between 1996 and 2008.

Large RMSE values at the tide records before 2008 may be explained by uncertainty in the model bathymetry, a shift in gauge
location, or inaccuracy of measurements caused by older, non-digital measuring instruments. As the quality improves from
2009 to 2015, the assumption that the measurement and / or the quality of the model bathymetry have improved seems most
likely. In 2000, an outlier value at CUX is observed which likely results from measurement errors.

After the tidal signal has been validated for its amplitude (i.e. tidal range), the vertical extent of the signal is checked by
controlling the error of the tidal high water to quantify bias. The comparison in Figure 11 is structured analogous to the tidal
345   range in Figure 10.

The RMSE distribution of the tidal high water shows RMSE margins between 5 cm in HEL and 20 cm in BAL. Most RMSE
values range in between 7 and 13 cm. The gauges NOY, HOO, ALW, HEL, BKA and HOH show RMSEs below 10 cm except
for NOY between 2004 and 2006. The largest RMSEs are computed in DUK, DWG and CUX which are all tide records located
in the mouths of the estuaries of Ems, Weser, and Elbe, indicating that the error in tidal high water increases upstream of the
350   outer estuaries. This is possibly related to insufficient horizontal and vertical grid resolution of the numerical model in the
complex bathymetry in the estuaries of the German Bight. Additionally, the gauge DWG suffers from systematic bias of 5 to
8 cm throughout all years which may also amplify its RMSE over proportionally.

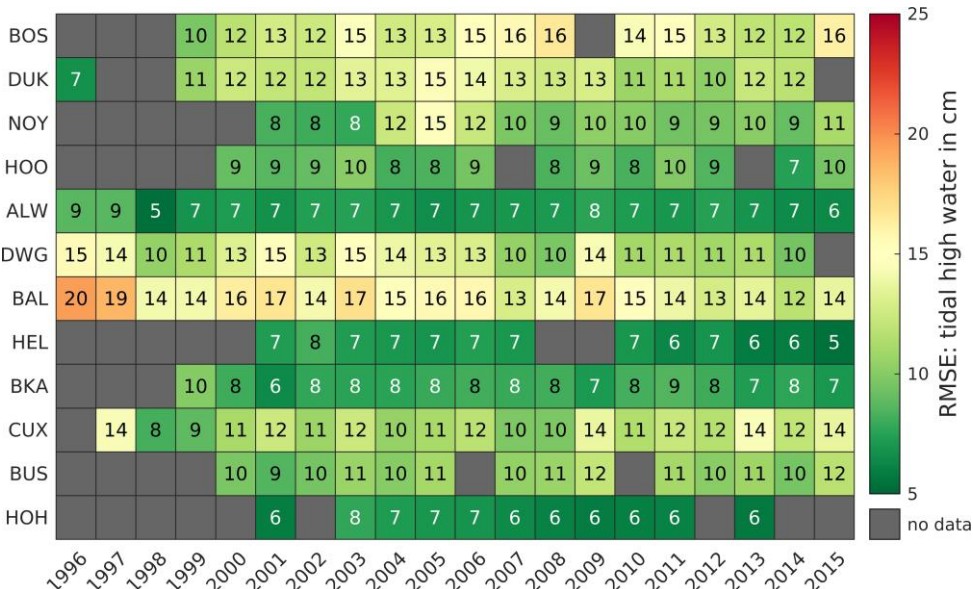

**Figure 11: Root mean square error (RMSE) of the tidal high water in cm between 1996 and 2015 at representative gauges.**

The flood duration is chosen as an indicator to verify the shape of the modelled tidal signal (also asymmetry or tidal distortion).
Figure 12 shows that the RMSE of the flood duration is between 10 and 20 minutes at most gauges. BKA, BUS and HOH
deviate with an RMSE of 20 to 37 minutes. While BKA and HOH show a constant deviation between 17 to 22 and 20 to 25
minutes, respectively, the deviation of modelled and observed data at BUS increases with time. After 2010 the error remains
constantly above 29 minutes which may result from local bathymetric changes that are not represented in model bathymetry,



especially in the morphologically active Meldorfer Bucht near BUS. This is likely, as topography influences tidal asymmetry strongly (Friedrichs and Aubrey, 1988).

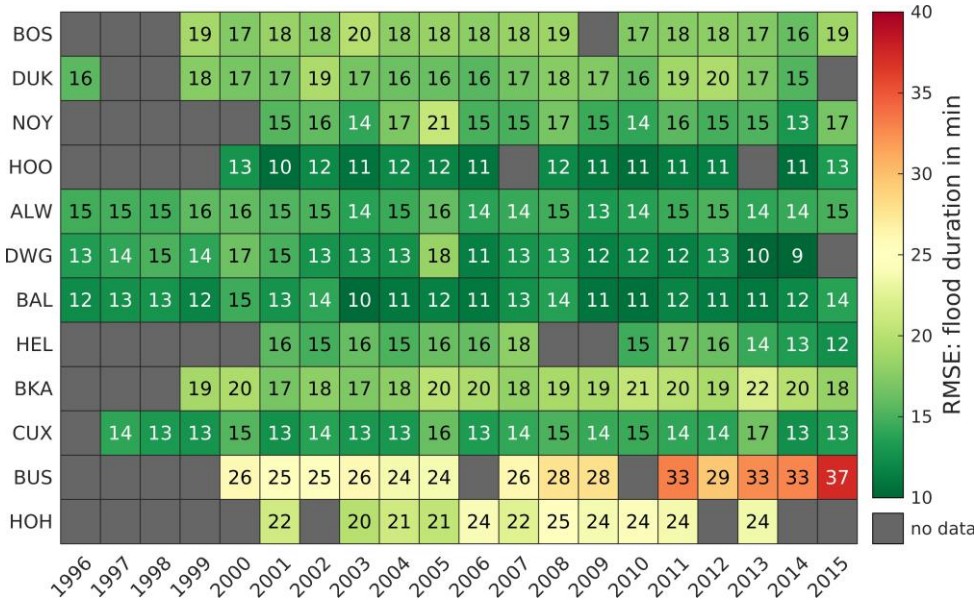

**Figure 12: Root mean square error (RMSE) of the flood duration in minutes between 1996 and 2015 at representative gauges.**

### 4.2 Current Velocity

The uncertainty of current velocity measurements (van Rijn et al., 2000), as well as the sensitivity of computed current velocities to water depth and water depth gradients have a strong impact which limits the applicability of a comparison between observed and modelled current velocity at individual locations. Nevertheless, we have carried out a validation of current
velocity magnitude for available data in the Ems, Elbe, and Jade estuary (Figure 13) with a statistical approach, to account for the limited possibility of a direct comparison. For validation, model data have been extracted at the depths of measurement devices. Samples are colored in the plots according to sample density. Moreover, the index of agreement R² and a linear regression with the slope *m* and the y-intercept *b* are used to obtain information about bias or time lag. The y-intercept *b* is an indicator for bias and the slope *m* for potential phase lag (Winter, 2007).

R² varies between 0.49 and 0.89, indicating high correlation between the predicted and observed current velocity magnitude. Comparisons at LZ1 and LZ4 show R² values of less than 0.51 which is due to a wider spread in the measured velocity data in the Elbe river in the year 2012. Thus, the regression parameters demonstrate cases of poor agreement as well. Model skill in the Ems and Jade estuary (a-c, f) however indicates strong agreement between prediction and observation, though low regression slopes below 0.77 are found in So07JD0, LZ1, LZ4 and KNO indicating a slight offset in the velocity signal between
flood and ebb.

For more information on the direction of current velocities, hodographs (provided in supplemental material S2) of the current velocity at LZ1 show that measured current velocity varies more in north- and southward direction, even though most of the ebb and flood peak currents are well reproduced by the model. The hodograph in LZ4 reveals that the model underestimates the ebb and flood current significantly as well as the cross-channel velocity variation. This is likely related to strong three-

dimensional effects at this location and issues with measurement quality at high current velocity magnitudes.

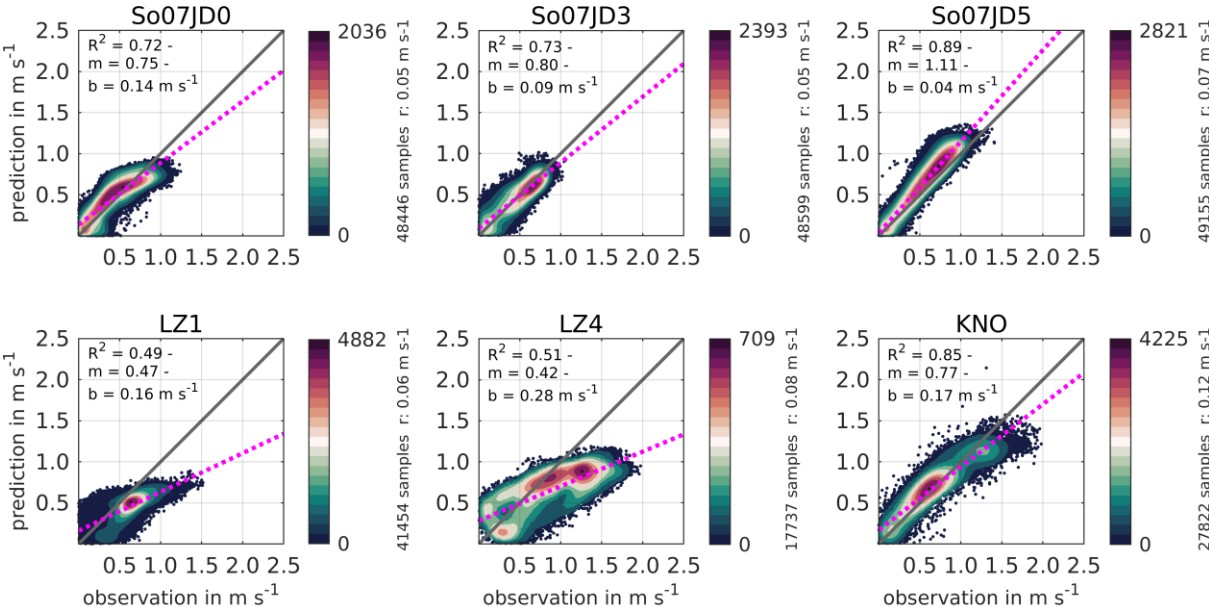

**Figure 13: Scatter plots of current velocity magnitude at the gauges in the Jade So07JD0 (a), So07JD3 (b), So07JD5 (c), the Elbe in LZ1 (d) and LZ4 (e) and the Ems in KNO (f) in 2012 in the German Bight. Figures are colored by sample density and contain the index of agreement R² as a measure for regression quality and the linear regression slope m and y-intercept b in m/s. The dotted pink line represents the linear regression and the solid black line and optimal correlation between observation and prediction.**

### 4.3    Salinity

Salinity is validated at gauges between 1997 and 2015 analogous to Sect. 4.1. The year 1996 was neglected due to the absence of observational data. We apply the RMSE for the observed and predicted salinity in Figure 14. No-data values result from

limited measurement availability to the authors, inconsistent data quality and quantity, or bias in the observational data sets. We focused on estuarine gauges as these demonstrate the highest salinity variation due to varying freshwater discharge. Nevertheless, a solid spatial and temporal coverage could be achieved in the German Bight. Note, that error margins for the RMSE of salinity depend on the amplitude of the salinity fluctuation during a tidal cycle, which is why RMSEs are typically lower outside of estuary brackish water zones.

RMSE values in Figure 14 vary between 0.7 and 4.5 ppt, though most RMSEs are in between 1 and 3 ppt. The best overall agreement is found in NUF and ALW, which are situated in the inner and outer Weser estuary, while the worst agreement is found in KNO (in 2000) in the Ems estuary and in LZ4 in the mouth of the Elbe estuary. Nearby gauges (e.g. LZ3 or LZ4a),





nevertheless, demonstrate lower RMSE values, which makes a slight vertical or horizontal misplacement in the model for LZ4 likely.


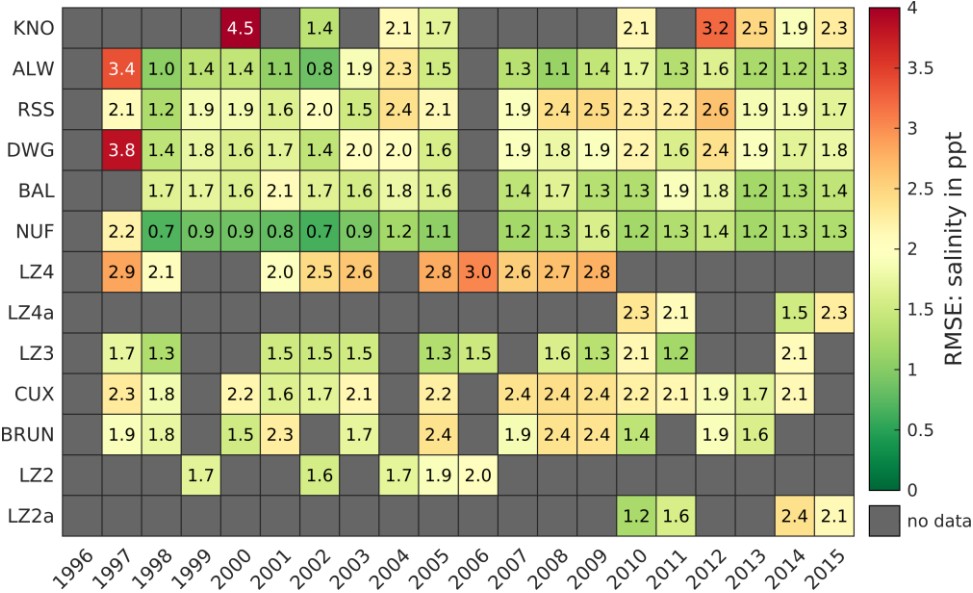

**Figure 14: Root mean square error (RMSE) of the salinity in ppt between 1996 and 2015 at representative gauges.**

### 4.4 Waves

We compare SWAN and UnK wave model results (significant wave height $H_{m0}$, mean wave period $T_{m02}$, peak period $T_p$ and
mean wave direction $\Theta_m$) against wave measurements in the German Bight by computing the RMSE. Wave measurements in the EasyGSH-DB product zone (EPZ) suffer from low data availability in contrast to, e.g. water level measurements. Most wave measurements in the EPZ were recorded in short-term measuring campaigns with a temporal extent of a few years at best. Hence, we decided to assess model performance for the product time span at a few locations only. Due to data gaps in measurements, it is impossible to validate every model time step within a year, which is why we provide an annual
completeness of the measured significant wave height in Table 2. Completeness hereby assesses the number of valid measurement samples at model output times divided by the total number of model output times. Measurements were checked visually for credibility and outliers and suspect measurement points were deleted. As mentioned before in Sect. 2.2, local water levels and current interaction are neglected for SWAN simulations. Thus, the validity of these results is limited to deep water conditions (i.e. areas with water depths of $\geq 20$ m).
Table 2 outlines wave validation results for a chosen year 2007 at the stations FINO1, Sylt, Elbe, and NSB-II (see Figure 9). All stations show limited completeness, between 89 % at Elbe and only 25 % at NSB-II. Deep sea measurements (FINO1, NSB-II) demonstrate lower RMSE values with SWAN, while nearshore samples (e.g. Sylt, Elbe) show better agreement from the two-way wave-current coupling of UnTRIM and UnK. Both models represent the significant wave height well at all stations





with a maximum RMSE of 0.74 in NSB-II (UnK). The RMSE of $T_{m02}$ ad $T_P$ remains within 3.22 s for both approaches, though
SWAN demonstrates a lower RMSE for wave periods. Mean wave direction displays RMSE values between 37.9 and 54.4 °,
although it must be noted, that mean wave directions from simulation results are compared with measured wave direction at
the peak frequency. These two values differ episodically, which is a likely explanation for low model skill. Analogous to the
RMSE of significant wave heights, larger deviations are notable whenever intermediate water depths (e.g. Sylt in Table 2) are
compared due to the applied mean water level and lacking current interaction in SWAN modeling approach which results in
inaccurate nearshore wave refraction.

**Table 2: Root mean square error (RMSE) of the significant wave height in meters ($H_{m0}$), mean wave period ($T_{m02}$), peak period - both in seconds ($T_p$), mean wave direction in degree ($\Theta_m$), water depth in m (d), and completeness of the measured significant wave height at selected locations in percent for the year 2007**

| Location | completeness in % | d in mNHN | RMSE $H_{m0}$ in m | | RMSE $T_{m02}$ in s | | RMSE $T_p$ in s | | RMSE $\Theta_m$ in deg | |
|---|---|---|---|---|---|---|---|---|---|---|
| | | | SWAN | UnK | SWAN | UnK | SWAN | UnK | SWAN | UnK |
| FINO1 | 70 | 29 | 0.29 | 0.61 | 1.11 | 1.43 | 2.50 | 2.66 | 40.7 | 47.7 |
| Sylt | 87 | 13 | 0.56 | 0.30 | 1.26 | 1.41 | 3.12 | 3.22 | 47.8 | 54.4 |
| Elbe | 89 | 25 | 0.20 | 0.40 | 1.10 | 1.27 | 1.53 | 1.76 | 40.6 | 50.2 |
| NSB-II | 25 | 44 | 0.39 | 0.74 | 0.86 | 1.18 | 2.14 | 2.27 | 37.9 | 42.5 |

Table 3 shows an assessment of the simulated significant wave height, mean wave period, peak period and mean wave direction
and available measurements at FINO1 (open sea research platform ca. 45 kilometers to the north of the East Frisian Island of
Borkum, see location in Figure 9; operational since July 2003) for the years 2003-2015. Analogous results at Elbe and NSB-
II are provided in the supplementary material S3 and S4 for the sake of completeness.

The annual RMSE of the significant wave height near the location of FINO1 ranges between 0.24 m and 0.32 m for the SWAN
and between 0.47 m and 0.66 m for the UnK model results. RSME-values are in the same order for other locations in deeper
water. Nevertheless, slightly differing RMSE values are observed for the locations Elbe and NSB-II, as larger deviations occur
in areas with intermediate water depths (e.g. RMSE near the location Sylt in Table 2) due to current interaction which leads to
better skill in the fully coupled UnTRIM2-UnK simulations. The mean wave period is systematically underestimated in the
both wave simulations. This phenomenon is known from comparisons of measured and calculated wave spectra from wave
hindcasts in the western Baltic Sea (Schlamkow and Fröhle, 2008) and is related to the wind energy input formulation applied
in SWAN. The annual RMSE of the mean wave period at FINO1 ranges between 1.07 s and 1.33 s (SWAN), and between
1.35 s and 1.61 s (UnK), respectively. The RMSE of the peak period at FINO1 varies between 2.07 s and 2.93 s (SWAN) and
between 2.14 s and 2.97 s (UnK). Differences >12 s between the observed peak periods are present, which can be explained
by differences between the observed and applied wind field over the North Sea. Moreover, it remains questionable how reliable
measured peak periods >12 s from a directional wave rider buoy, such as FINO1, are. Hence, outlier differences of the peak



period may be related to coarse measurement resolution of long wave periods and wave spectra. The annual RMSE of the mean wave direction at FINO1 is between 33.7° and 40.7° (SWAN) and between 40.4° and 47.7° (UnK), due to the reasons explained above.

**Table 3: RMSE of significant wave height ($H_{m0}$), mean wave period ($T_{m02}$), peak period ($T_p$), mean wave direction ($\Theta_m$) and**
**measured $H_{m0}$ completeness from 2003 to 2015. Note, that FINO1 started operating in July, 2003, which is why no RMSEs could be calculated before.**

| Year | completeness $H_{m0}$ in % | RMSE $H_{m0}$ in m | | RMSE $T_{m02}$ in s | | RMSE $T_p$ in s | | RMSE $\Theta_m$ in deg | |
|---|---|---|---|---|---|---|---|---|---|
| | | SWAN | UnK | SWAN | UnK | SWAN | UnK | SWAN | UnK |
| 2003 | 23 | 0.32 | 0.66 | 1.21 | 1.61 | 2.07 | 2.20 | 34.0 | 42.2 |
| 2004 | 52 | 0.28 | 0.55 | 1.31 | 1.58 | 2.18 | 2.19 | 33.7 | 40.4 |
| 2005 | 89 | 0.28 | 0.56 | 1.26 | 1.61 | 2.01 | 2.14 | 36.6 | 42.7 |
| 2006 | 63 | 0.28 | 0.54 | 1.07 | 1.35 | 2.53 | 2.61 | 39.8 | 46.7 |
| 2007 | 70 | 0.29 | 0.61 | 1.11 | 1.43 | 2.50 | 2.66 | 40.7 | 47.7 |
| 2008 | 79 | 0.28 | 0.60 | 1.33 | 1.60 | 2.68 | 2.76 | 38.6 | 45.5 |
| 2009 | 42 | 0.24 | 0.47 | 1.15 | 1.38 | 2.58 | 2.70 | 38.7 | 46.5 |
| 2010 | 63 | 0.26 | 0.58 | 1.16 | 1.45 | 2.11 | 2.24 | 40.4 | 45.1 |
| 2011 | 91 | 0.26 | 0.59 | 1.19 | 1.47 | 2.4 | 2.74 | 37.6 | 45.6 |
| 2012 | 40 | 0.28 | 0.62 | 1.12 | 1.44 | 2.43 | 2.52 | 35.6 | 41.9 |
| 2013 | 97 | 0.26 | 0.57 | 1.22 | 1.49 | 2.49 | 2.58 | 32.9 | 39.2 |
| 2014 | 69 | 0.26 | 0.54 | 1.24 | 1.46 | 2.93 | 2.97 | 34.1 | 42.2 |
| 2015 | 90 | 0.28 | 0.61 | 1.06 | 1.41 | 2.64 | 2.73 | 32.7 | 41.1 |

## 5 Data availability

Open-access EasyGSH-DB data products can be obtained separately in two categories for hydrodynamic analyses (10.48437/02.2020.K2.7000.0003) as GeoTIFF / ESRI shape files and hydrodynamic simulation results
(10.48437/02.2020.K2.7000.0004) in a common structured NetCDF format. Stationary wave products count within the simulation results category and are available in an ASCII format for further processing or direct nesting. EasyGSH-DB data can be obtained by download and via web service (YYYY translates to one year between 1996 to 2015):

- web map service (WMS, http://mdi-dienste.baw.de/geoserver/EasyGSH_Kennwerte_**YYYY**/wms),
- web feature service (WFS, http://mdi-dienste.baw.de/geoserver/EasyGSH_Kennwerte_**YYYY**/wfs)
- web coverage service (WCS, http://mdi-dienste.baw.de/geoserver/EasyGSH_Kennwerte_**YYYY**/wcs).





An overview of products, publications and web services can be found on the EasyGSH-DB website (https://easygsh-db.org, last access: January 7th 2021). Users can view, animate, and explore data through interactive web map viewers. All data underly the Creative Commons license 4.0 (CC-BY 4.0).

## 6    Conclusions and future recommendations

The presented integrated, marine data collection for the German Bight from 1996 to 2015 establishes a reliable, high resolution data base of hydrographical parameters for scientific, commercial, and governmental organizations. Based on the involvement and participation of coastal stakeholders, hydrodynamic model results (i.e. sea surface elevation, current velocity, bed shear stress, salinity, wave parameters) are provided file-based and online on a 1,000 m grid with 20-minute time intervals in the German Bight. Additionally, analyses products (tidal characteristic values, e.g. tidal range, tidal high water, flood current

velocity, significant wave height) have been created from simulation results to improve the accessibility of this data set and to reduce the data size. Data products are extensively validated and can be used for various applications in oceanography, earth sciences and coastal engineering, though the limitations defined in this report must be considered before application.

  The numerical modelling approach aims to provide a synthesis of consistent, relevant forcing parameters (e.g. fresh water discharge, wind speed, and tidal dynamics) and geomorphology. Annually updated geomorphology is a unique feature of this

data collection compared to previous works which have mainly considered static bathymetry over short and long time spans. By using the same basic assumptions concerning grid configuration and -resolution, numerical parameters, friction height, surge assimilation, wind forcing as well as fresh water discharge for 20 years, this investigation produces a homogeneous consistent dataset. The collection can therefore be the starting point for more detailed simulations in the German Bight to further increase our understanding of the complex dynamic processes combining geomorphology and hydrodynamics. It should

be mentioned that data were created as input for the future of mobility, i.e. it is hoped that it contributes to potential application in green energy. From a scientific perspective, consistent, long-term tidal characteristic values are not a novel concept, but scientific practice has shown, that their application is advantageous, though they are not used commonly. In Sect. 4, we have touched upon its potential to describe ability of a model to reproduce main tidal properties of a large area over long time scales, using only a few tidal characteristic parameters.

Even though, the marine data collection already covers a time span of 20 years, the time span still is too short for many applications (e.g. mean sea level science). Thus, we propose to pursue a continuous extension of the data collections from 2016 onwards. An extension towards the past can also be performed, though it must be noted that the quality of input forcing data decrease drastically before 1996. Finally, we emphasize that any modelling approach critically depends on the availability of *international* field data, especially for the ever-changing bathymetry, measurements for validation and boundary as well as

initial forcing data. This stresses the immediate need for international mutual data bases, minimum quality standards, good scientific practice to reduce data clutter, and complete inspire-conform meta data.



## 7 Author Contribution

Robert Hagen – article composition, article figures, article concept, numerical modeling UnTRIM2- UnK, validation UnTRIM2, conceptual product design, lineage design

Andreas Plüß – project initiation, supervising, proof-reading, conceptual product design, lineage design

Romina Ihde – meta data and repository management, digital object identifier registration

Janina Freund – model result processing, tidal characteristic, and harmonic analysis

Norman Dreier – numerical modeling SWAN, model result processing and analysis (SWAN waves), validation SWAN, proof-reading

Edgar Nehlsen – supervising, proof-reading, lineage design

Nico Schrage – numerical modeling SWAN (grid-composition), model result processing and analysis (SWAN waves), lineage design

Peter Fröhle – project initiation, supervising, proof-reading

Frank Kösters – project initiation, supervising, proof-reading, article concept

## 510 8 Competing Interests

The authors declare, that they have no conflict of interest.

## 9 Acknowledgements

The authors thank the German Federal Ministry of Transport (BMVI) for funding the mFUND project EasyGSH-DB (funding no. 19F2004A), which has made this data collection possible. We also thank the suppliers of field measurements used to 515 validate our numerical models (i.e. supplement S1). We would like to express their gratitude to all EasyGSH-DB collaborators being smile Consult GmbH, Küste und Raum and the Federal Maritime and Hydrographic Agency, Germany (BSH) for their valuable input and enjoyable corroboration. We would also like to acknowledge the many stakeholders and beta-testers, which have provided constant feedback to us and contributed their time to improve our data products presented herein. The BAW authors furthermore wish to express their gratitude to the PROGHOME team for excellent software solutions and extraordinary 520 support at a very high level. RH and JF thank Günther Lang for his important input towards the description of the analysis methodology.



# 10   Appendix

## 10.1   Error Metrics

To describe the quality of a model, an error threshold must be specified. An error $E_t$ in model validation concerns the difference between observed (O) and predicted (P) values (or vice-versa). The mean error (*ME*, equation 1) is the arithmetic mean over the difference of observed and predicted values at location for *N* samples at mutual time *t*. The standard deviation $\sigma$ (equation 2) describes the error spread of the error distribution around the mean error.

$$ME = \frac{1}{N} \sum O - P = \frac{1}{N} \sum E_t \tag{1}$$

$$\sigma = \sqrt{\frac{1}{N-1} \sum |E_t - \mu|^2} \tag{2}$$

The root mean square error (*RMSE*, equation 3) takes the root of the mean squared errors $E_t$. The squaring of differences
weighs the *RMSE* towards larger error margins. Note that any information about over- or underestimation is lost, due to the application squared errors.

$$RMSE = \sqrt{\frac{1}{N} \sum (E_t)^2} \tag{3}$$

The index of agreement $R^2$ (equation 5) is defined as an indicator of how closely two data sets are related. The relationship is classified with the squared Pearson correlation coefficient, thus 1 represents perfect, 0 no and -1 an antagonized relationship between two data sets.

$$R^2 = \sqrt{\frac{\frac{1}{N} \sum (P_t - \bar{P})(O_t - \bar{O})}{\sigma_P \sigma_O}} \tag{5}$$




## 10.2 Product List

**Table A1: Model simulation results**

| data product | zone | unit | interval | resolution |
|---|---|---|---|---|
| sea surface elevation | EPZ | m | 20-minute | 1,000 m |
| current velocity (eastward, northward) | EEZ | m/s | 20-minute | 1,000 m |
| salinity | EPZ | ppt | 20-minute | 1,000 m |
| bed shear stress | EEZ | N/m² | 20-minute | 1,000 m |
| waves (spatial) | EPZ | m | 20-minute | 1,000 m |
| 1d and 2d wave spectra | station | - | 20-minute | local |
| spectral wave parameters | station | - | 20-minute | local |

**Table A2: Wave analysis products divided by the simulation software UnK and SWAN**

| data product | zone | variant | unit | interval | resolution |
|---|---|---|---|---|---|
| significant wave height (UnK) | EPZ | 50-, 95-, 99% quantile, max. | m | annual | 100 m |
| mean wave period ($T_{m02}$) at max. significant wave height (UnK) | EPZ | mean | s | annual | 100 m |
| directional energy density spectrum (SWAN) | station | - | m²/Hz | annual | point |
| significant wave height $H_{m0}$ (SWAN) | EPZ | mean, 50-, 95-, 99% quantile | m | annual | 100 m |
| peak period (SWAN) | EPZ | mean, 5-, 50-, 95 % quantile | m | annual | 100 m |
| mean wave direction | EPZ | energy-weighed | ° | annual | 100 m |
| wave energy (SWAN) | EPZ | mean | Ws/m² | annual | 100 m |




**Table A3: Tidal characteristic and harmonic data analysis products**

| data product | zone | variant | unit | interval | resolution |
|---|---|---|---|---|---|
| high tide | EPZ | 5-, 50-, 95 % quantile | m | annual | 100 m |
| high tide | EEZ | 5-, 50-, 95 % quantile | m | annual | 1,000 m |
| low tide | EPZ | 5-, 50-, 95 % quantile | m | annual | 100 m |
| low tide | EEZ | 5-, 50-, 95 % quantile | m | annual | 1,000 m |
| tidal range | EPZ | 5-, 50-, 95 % quantile | m | annual | 100 m |
| tidal range | EEZ | 5-, 50-, 95 % quantile | m | annual | 1,000 m |
| mean tide | EPZ | 50 % quantile | m | annual | 100 m |
| mean tide | EEZ | 50 % quantile | m | annual | 1,000 m |
| number of high tide events | 12-SM | total number | - | annual | 100 m |
| number of low tide events | 12-SM | total number | - | annual | 100 m |
| mean inundation period | 12-SM | mean | min | annual | 100 m |
| mean flood current velocity | EPZ | mean (magnitude, x, y) | m/s | annual | 100 m |
| peak flood current velocity | EPZ | 5-, 50-, 95 % quantile | m/s | annual | 100 m |
| mean ebb current velocity | EPZ | mean (magnitude, x, y) | m/s | annual | 100 m |
| peak ebb current velocity | EPZ | 5-, 50-, 95 % quantile | m/s | annual | 100 m |
| ratio of mean flood to mean tide current velocity | EPZ | mean | - | annual | 100 m |
| ratio of mean ebb to mean tide current velocity | EPZ | mean | - | annual | 100 m |
| mean salinity per tide (annual mean) | EPZ | 5-, 50-, 95 % quantile | ppt | annual | 100 m |
| peak bed shear stress during flood | EPZ | 50-, 95% quantile | N/m² | annual | 100 m |
| peak bed shear stress during ebb | EPZ | 50-, 95% quantile | N/m² | annual | 100 m |
| mean bed shear stress during flood | EPZ | mean (x, y) | N/m² | annual | 100 m |
| mean bed shear stress during ebb | EPZ | mean (x, y) | N/m² | annual | 100 m |
| sea surface elevation | EPZ | 1-, 99 % quantile | m | annual | 100 m |
| salinity | EPZ | 1-, 99 % quantile | m | annual | 100 m |
| M2 amplitude | EPZ | - | m | annual | 100 m |
| M2 phase | EPZ | - | ° | annual | 100 m |



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
