# Peer review of "An Integrated Marine Data Collection for the German Bight – Part II: Tides, Salinity and Waves (1996 – 2015)"

_Earth System Science Data, 2021_

## Author Comment (AC1)

Dear anonymous ESSD reviewer,

first, thank you for reviewing our manuscript for ESSD. We appreciate your voluntary work during pandemic times which enables us to still publish research and make the EasyGSH-DB data collection available to the public. Let us answer to your remarks below.

Sincerely,

Robert Hagen (on behalf of the authors)

**Answers:**

The manuscript provides a thorough and detailed overview of the process of collection and generation of a large data set of tides, salinity and waves for the German Bight. The efforts towards systematization, validation and gridding of the data set are substantial and professional, and have led to a multi-purpose data base that can be used for a variety of different tasks. The time interval (20 years) is shorter than the time period used for quantification of the climate (30 years) but hopefully the data set will be extended to fully cover at least one classic climatological time period (e.g., 1991–2020).

I am in favour of publication of this paper and only recommend adding a few remarks and adjusting several minor items.

1) While the data about tides and salinity match the relevant measurements well, the deviations of the measured wave properties from the hindcast ones are fairly large. Some shortages (part of the systematic mismatch of hindcast and measured wave periods) seem to stem from model-specific properties.

   Answer: A systematic mismatch between observed and calculated mean wave periods were related to the calculated wave spectra by SWAN in the past (beginning of Sect. 4.4 in the manuscript). Larger differences of the peak wave period may arise from the model set-up, which neglects the wave boundary conditions at the open boundaries to the North Atlantic as you have stated.

   Action: We revised some statements in Sect. 4.4 in the revised manuscript towards neglecting open boundary wave conditions and implications.

2) As the K-model does not involve nonlinear interactions of waves, it is natural that it underestimates the wave periods in some occasions. It is essentially a coastal model and works well in situations where most of the wave fields are relatively young wind seas. The inability of the SWAN model to capture some wave periods is somewhat more intriguing and should be commented separately. I guess that this feature is unavoidable as swells generated by storms between Norway and Island may easily reach the German Bight, and they are not captured by the particular model set-up. The discussed feature does not diminish the value of the paper and the underlying data sets but I would still recommend to make clear in the conclusions that the quality of some parts of the data set of wave properties is lower than that of the majority of this pool of data, and to indicate the reasons.

   Answer: We agree with your point that model limitations towards swell waves from the Northern Atlantic should be emphasized more. In addition to our revision from 1), we gave some more remarks on the ability of SWAN and UnK to capture mean and peak wave periods. Moreover, considering the comments on the limitation of the quality of the SWAN simulation results

(see Lines 417-419 in the original manuscript), we agree that swell wave events are not fully captured by the particular model set-up due to missing wave boundary conditions along the open model boundaries to the North Atlantic which might lead e.g. to an underestimation of peak wave periods during calm-weather conditions. We have also elaborated more on the differences between the two wave models.

Action: We added additional information on that issue in the beginning of Sect 4.4 in the revised manuscript. Additional information on the differences between the wave models was added at the end of Sect. 2.2 and at the end of Sect. 4.4 above Table 3.

Minor issues:

Generally, all abbreviations should be explained at their first appearance.

3)  Line 35: consider saying "spatially varying tidal range" instead of "spatially varying increase and decrease in tidal range".

    Answer: The spatially varying increase and decrease concerns spatially variable change rates of tidal range, not the spatial variability of tidal range itself.

    Action: Changed "increase and decrease in" to "change of"

4)  Line 45: probably ERA-40 is meant.

    Action: Changed to "ERA-40" in the manuscript.

5)  Line 68: explain FES.

    Action: We added a definition and a short description to the manuscript.

6)  Line 90: it is better to explain also CSV.

    Action: Changed in the manuscript.

7)  Line 91: explain THREDDS or give a link address.

    Action: We have added an explanatory link.

8)  Line 97: TM1 and TM2 are probably the same quantities as described on page 11, line 254. Please unify.

    Action: Changed in the manuscript.

9) Line 135: explain BAW.

Action: We added an parenthesis to the reference, as BAW in this case is part of a reference.

10) Line 228-229 and elsewhere: perhaps it makes sense to use g/kg instead of ppt or at least say that g/kg is today a standard unit.

Action: As the unit 1e-3 (i.e. ppt) is documented within our salinity product data, we suggest to keep "ppt" in the text. We acknowledge your point by adding (i.e. g/kg) in the beginning of Sect. 3.1 after the first mention of ppt.

11) Line 265: "The annually, tidally averaged salinity" sounds strange; please clarify.

Action: We agree and have removed the "annually" because it is mentioned several times in the manuscript that analyses products are annually averaged results.

12) Line 294: the last symbol in UnTRIM2 should be a superscript; also on lines 443, 498, 499.

Action: Changed in the manuscript (no track changes).

13) Line 315, caption to Table 1: it is recommended to use "degrees" instead of the degree sign

Action: Changed in the manuscript.

14) Line 423: here it is only UnTRIM, with no "2" at all; please unify.

Action: Changed in the manuscript (no track changes).

15) Line 448: "Differences >12 s between the observed peak periods are present…" is ambiguous; please specify whether peak periods over 12 s are meant or is the difference between the observed and modelled peak periods that exceeds 12 s. This may of course happen for extremely long-period swells that are not resolved by the model.

Answer: This sentence was revised completely due to the remarks of reviewer #2. For this reason, we have not conducted specific changes for this point.

16) Line 461: simply "in ASCII format".

Action: Changed in the manuscript.

17) Line 533: "squared" would render the resulting value of the Pearson correlation coefficient into the range [0, 1] and make perfect and antagonistic correlations indistinguishable; is this what you mean? Anyway, I guess something is wrong with Eq. (5) as the expression under the square root can easily be negative and the entire right hand side of this equation expresses square root of the Pearson correlation coefficient. Also, Eq. (4) is missing.

Answer: There has been a wrong wording in Appendix 10.1. We did not mean to mention the index of agreement $d$, but the coefficient of determination $R^2$ which does fit into a range [0, 1]. After rechecking our text, however, we do not believe Eq. (5) to be essential, as it is a well-known parameter for the goodness-of-fit in regression /curve-fitting and has only been used once in Section 4.2.

Action: We have reworded the last paragraph in 10.1 to a simple description of $R^2$ and removed Eq. (5).

18) References: Some titles are fully capitalized, some not. Several references are incomplete. For example, volume number is missing in (Battjes and Janssen), Jänicke et al. (2020) seems incomplete, Janssen et al. (1999) is distorted, the number of pages of (Kösters et al., 2014) and especially (Plüss, 2003) is surprising, Müller (2011) misses some data, it should be "height" in van Rijn et al. (2000), and Winter (2011) is incomplete.

Action: We have adjusted references as follows (no track changes):

- Some article titles are capitalized depending on the journals policy at the time. JGR: Oceans for example has changed their spelling to capitals after 2017. We have changed our citation style to "Copernicus_Publications [As of 2019]" (as required by ESSD) which unified some capitalization in the references. Note, that some German titles still show additional capitalization as this is required by German spelling. I would like to leave further design decisions concerning capitalization with ESSD editing, as their example section contains both, capitalized and non-capitalized reference titles.
- Battjes and Janssen (1978): Added a 1978 behind Coastal Engineering, as the conference, not the journal is referred to.
- Jänicke et al (2020): Changed reference type to article, which fixed the issue
- Janssen et al. (1999): Removed the spare DOI
- Kösters et al. (2014): Changed this source to the AufMod project synthesis, which references all AufMod results alike: Heyer, H., Schrottke, K., Zeiler, M., and Plüß, A.: Synthese der interdisziplinären Forschung in AufMod, Die Küste, 181–191, 2015.
- Plüß (2003): Added the correct citation information
- Müller (2011): Removed page numbers (not necessary here)
- van Rijn et al. (2000): added missing "h".
- Winter (2011): We are unsure how this title is incomplete. Here is the citation suggested by JCR: "Winter, C. (2011). Macro scale morphodynamics of the German North Sea coast. *Journal of Coastal Research,* 706-710. Retrieved March 24, 2021, from https://www.jstor.org/stable/26482263". We have added "(retrieved March 24, 2021) at the end of our citation. Since this was a conference proceedings issue, there is no volume number or DOI available.

---

## Author Comment (AC2)

Dear anonymous ESSD reviewer,

first, thank you for the thorough review of our manuscript for ESSD. We acknowledge that you made some in depth comments which will improve our manuscripts accessibility. We will try to address your points in detail. Let us answer to your remarks below point-by-point.

Sincerely,

Robert Hagen (on behalf of the authors)

**Answers to Major Comments:**

1. Compared to earlier hindcasts this study uses time-dependent (annual) bathymetric data, which represents one of the main novelties. At the same time, statements are made that errors in earlier years are caused by uncertainties in earlier bathymetric data. This left me with two questions: First, how have these data been derived? Second, is there value-added compared to simulations with constant bathymetry? More specifically, I suggest that the authors:

   (a) introduce some description of the bathymetric data use, for example, the extent to which the model domain was surveyed every year or how much time interpolation was done at individual grid points, or how good different surveys match spatially and whether or not interpolation was needed. Some Figures to illustrate the issues would be appreciated.
   (b) provide a comparison of results with a run with constant bathymetry to assess the value added by their approach. I am fully aware that the hindcast could not be fully repeated, but some strategy, e.g. to assess the sensitivity for average or extreme conditions would be appreciated.

   Answer to (a): We recognize that a clearer reference to the first part of our data publication is necessary as many of your justified concerns towards data reliability are addressed in Part I. An actual validation of our bathymetries is hardly possible because of the large area (several 1,000 km² in the Wadden Sea alone) and numerous authorities with different measurement techniques which is why we perform a plausibility assessment in Sect. 4 (Part I) for the bathymetry in 2016. Herein, the e.g. data density (Sect. 4.1) and data sources (Sect. 4.2) for each geomorphological data product are documented. As seen in Fig. 11, a (Part I), data density declines rapidly after the shallow, near-shore areas and has varying input resolution between more than 200 m to less than 1 m. Fig. 11, b (Part I) illustrates an exemplary data source map which distinguishes between survey type (measurement technique), survey time, authority, etc. Data source maps are always provided twice, as the interpolation in time always interpolates between the closest data set in the past and future. Obviously, survey quality and frequency have improved over time which is why we included the statement of uncertainties in earlier data. These uncertainties are enhanced by the fact that very precise and high-resolution laser scan data have become available between 1999 to 2005 in the EPZ. All data density and data source maps are distributed with our bathymetry products.

   Answer to (b): In L58-61 (Sect. 1) we have referenced the work of Jacob et al. (2016) who have performed such a sensitivity study using 2000 and 2011 AufMod bathymetry. They showed implications on 2M2-M4 phase lag, and tidal range in the entire German Bight. I hope you understand that adding a sensitivity study on this subject would be a topic for a paper on its own.

   Action: We have clarified the scope of the two parts of our joined publication with Sievers et al. (2021, under review) in the introduction (also see major point 8.). We have added the data citation to the bathymetric data set to the caption of Figure 4.

2. The domain of the hydrodynamic model basically covers the North Sea and the authors briefly mention that some calibration was made to account for surges at the open boundaries. Some more details are needed, for example, what data were used and how was the calibration made? Can this set-up account for the effects of external surges? What are the impacts on extremes?

Answer: We suppose this refers to L131-134 of the original manuscript. The external surge is calculated from differences between calibrated hydrodynamics and sea surface measurements. Let us take you through the external surge assimilation process as documented in Plüß (2003):

(1) Hydrodynamic calibration of the model
(2) Calculation of differences between model result and measurement near open boundary
(3) Smoothing of those differences by running average
(4) Adding the smoothed differences onto the open-boundary water level signal
(5) Re-running the model
(6) Repeating step (2-5), until satisfying agreement is reached or differences are near zero.

We used tide gauges from Roscoff (France) for the southern open boundary, and Helgoland (Germany) for the northern open boundary. We are aware that Helgoland is quite far away from the northern boundary but our model simulations showed much better agreement to measurements with Helgoland than with Lerwick or Aberdeen (both Scotland), similar to the results of Plüß (2003) and Heyer (2015). Also, sea surface measurements in Lerwick and Aberdeen suffer from numerous data gaps throughout the product time-span which makes their application quite cumbersome. External surge calculated in Helgoland and Roscoff includes a time-lag corresponding to the average propagation speed of the tidal wave. The entire process needs to be repeated 2 to 7 times each year to catch smaller surge fluctuations and to improve extreme events.

To answer your questions: We expect that our setup fully includes external surge for fair-weather and extreme conditions alike (also see our answer to 3). We believe the model calibration process to be redundant in this paper, as we want to focus on the data product validation rather than modelling. Many aspects of the calibration (e.g. parametrization or bottom roughness) are documented seperately in BAW Technische Berichte (2020) which can be downloaded from the EasyGSH-DB website (or via DOI: 10.18451/k2_easygsh_1).

Action: We added additional information about the external surge approximation process and iteration as well as the benefits to our data in the beginning of Sect. 2.2 (around L135). Specifically, we added that surge is calculated separately at the northern and southern open boundary by smoothed water level differences, and that we reiterate surge, until these smoothed differences are near zero. We have also elaborated on the benefits of this procedure, as sea level rise, external surge, and every other water level fluctuation is considered in our data.

3. The validation strategy appears a little limited. While it somehow depends on how the data will be used (the manuscript is not very specific here) I feel that extreme events may play a role. Most of the validation concentrates on average errors, some specific validation for extremes would be very beneficial.

Answer: Thank you for that helpful comment, the missing validation of extreme events was a shortcoming in our validation strategy, which we have added now. We chose to extend our validation methodology in chapter 4.1 by adding the error of the 99 % quantile of sea surface (which is an annual data product in the data collection). Figure 13 shows that the model reproduces extreme events well in the range of centimeters to tens of centimeters with declining accuracy towards estuaries, as to be expected.

Action: We have included an additional heatmap (i.e. Figure 13, see below) showing the error of the 99 % water level quantile at representative gauges in each year analogous to the remaining water level validation in Sect. 4.1 and added some descriptive text.

[Figure]

4. I had difficulties following the argumentation in section 2.1. From my perspective, this section requires major re-writing. The envisaged intention appears not to be in-line with the discussion. Figures 1 and 2 need better description and the general strategy needs to be explained. For example, the text discusses web and GIS-based applications that are not apparent in the scheme in Figure 1. I was more or less lost reading this section. Are the model outputs part of the data set or only the analyzed products? What were the criteria for selection of specific analyses (user request, other, ...), etc.

Answer: We agree with your view, and have completely restructured Sect. 2.1. We believe that it was confusing to explain the product lineage before introducing products themselves. We have now paid special attention to repeatedly use our definitions from Figure 2 (former Figure 1) in the text as often as possible and added a definition on how data products differ from model- and analysis, respectively. We have also tried to introduce the figures more prominently in Sect. 2.1 as you suggested.

Action: We have now introduced data products and product areas first by switching Figure 1 and 2. Then we start describing product heritage, metadata assets, and lineage. Figure 1 and 2 have also received more text and their caption has been sharpened. The emphasis on GIS applications has been removed, as this is not the focus of the Section.

5. The experimental design is not fully clear. The authors should discuss and justify the use of two different wave models. I understand that one was coupled to the hydrodynamic model but why then the other was needed? If there is a reason this should be elaborated.

Answer: We have deployed two different wave models, because both models have different characteristics and consider different physical processes. While we were able to run UnK online with sea surface and current, SWAN considers more detailed processes, such as triads, quads, white-capping, or wave breaking, which is not implemented in UnK. For this reason, we believe that it was

inevitable to assess the sea state with two models with each having its individual benefits and short-comings. Another important result of using two models was the comparison to measurements, as a similar error range solidifies the quality of sea state products in our data collection.

Action: We have modified the description of our wave setup at the end of Section 2.2. An additional sentence was added which clarifies, why the limited amount of wave processes in UnK is a short-coming which had to be addressed by a second model. We have also added an additional sentence which highlights that we hope to gain confidence in our data by applying two models at the end of Section 2.2. A suggestion to users was added at the end of Sect. 4.4 for which wave data to use in which case.

6. There is something wrong with the error metrics (equation #5 in the appendix). This is (a) not a typical definition of an index of agreement and (b) if a correlation is meant both sides of the equation do not match. Maybe this is just a typo, but please check potential consequences for analyses in the text. Also, there is no equation #4.

Answer: This is correct, there is a copy-paste error on our side in L535. We meant the coefficient of determination $R^2$ instead of the index of agreement $d$. We have decided to remove Eq. 5 in accordance to remarks of reviewer #1 as it is a well-known standard procedure for the goodness-of-fit in regression and is only applied once in Figure 14. A revised description of the parameter is still given.

Action: We removed Eq. 5 and revised its description.

7. The paper mentioned several times the involvement of stakeholders in the development of their product but their role or the benefit for the overall product is not addressed.

Answer: We agree that the stakeholder involvement is underrepresented in its complexity in the paper. We have attempted to acknowledge the participation of stakeholders by mentioning in the end of the introduction that their involvement has led to additional data products but we see now that this is too vague at this point. In EasyGSH-DB, we worked together with more than 10 different coastal stakeholders with applications varying from planning cable routes to offshore wind energy farms, the definition of intertidal areas with remote sensing, or the evaluation of bottom shear stress for sea weed seeding to name only a few. Although the stakeholder involvement was an important feature in the data collection design process, we believe that the presentation of our data should be most prominent in the paper.

Action: We have added an explanatory sentence towards the above-mentioned disciplines at the end of the introduction in Sect. 1 to strengthen our paper in the regard of stakeholder involvement without overstating their relevance for data products in this paper. Stakeholder involvement has also been removed from the abstract.

8. The paper mentions several times that it is the second part of a two-part publication. I suggest that the authors at the beginning briefly introduce the overall concept and how work/results were split into different parts. This would make the strategy and content easier to assess. The parts where this statement is repeated could then be removed.

Answer: We have decided to divide the data publication for the EasyGSH-DB data collection into a geomorphological and hydrodynamic component, as results from the first publication are

methodically completely separated from our work and a combined description effort would have resulted in an underrepresentation of either part. Considering the concerns you raised in point 1), we still believe this to be the correct decision, as the handling and time-interpolation of bathymetry over such a large area presents a novelty in itself. Hence, our answer to 8) mostly refers to the statements from point 1a). We believe that the additions we have made at the end of the introduction now cover the reasons why there have been two, instead of one paper.

Action: see answer to 1a), no specific action to 8)

**Answers to Minor Comments**

9. In the title, what is the meaning of the CE in "2015 CE"?

   Answer: CE stands for "Common Era" and defines that we are not discussing any historic data.

   Action: We decided to remove "CE" from the title.

10. The abstract should be re-written. It should not contain references and should describe the work presented in this manuscript.

    Answer: The references for our data and the data DOI are mandatory in the abstract in ESSD. We cite from the ESSD homepage:

    "Abstract: the abstract should be intelligible to the general reader without reference to the text. After a brief introduction of the topic, the summary recapitulates the key points of the article and mentions possible directions for prospective research. Reference citations should not be included in this section **(except for data sets)** and abbreviations should not be included without explanations. At least for the final accepted publication, a functional data set **DOI and its in-text citation** must be given in the abstract. If multiple data set DOIs are necessary, please instead refer to the data availability section."

    We furthermore believe that the abstract should represent the data products we have created instead of the technical manufacturing process. Therefore, we have tried to go from motivation to data set overview and novelty, before giving a brief description and an outlook. Nevertheless, we have made minor adjustments to the abstract due to some of your other comments (see below).

    In our understanding, the abstract should focus on the data products instead of the work necessary for their creation. Nevertheless, there have been many small adjustments at the abstract over the course of language editing and other some minor adjustments. We hope these changes address your concerns.

    Action: We added more precise description of the data (i.e. depth-averaged) and removed the stakeholder involvement (see major point #7, and minor point #18). Many small changes concerning wording have been made.

11. Line 27: To me, the phrase "in a contested region" sounds strange in this context. I suggest replacing with, for example, "… region with conflicting or competing interests …"

Action: Changed to "is a region with where competing interests of economic growth and the protection of future ecosystem services collide" in the manuscript.

12. Line 34: What exactly is "short-term" with the respect to the previous sentence?

Action: We removed "short-term" from the manuscript.

13. Line 37: "For this reason …". I don't think that is the reason.

Action: We have revised the sentence to "Thus, an unusually broad spectrum of field data and scientific knowledge is available at the study site, because hydrographic parameters of the North Sea are monitored by one of the densest measurement networks worldwide.".

14. Line 40: "Individual tides" Please be specific.

Action: Changed "individual tides" to "individual tidal events" in the manuscript.

15. Lines 41-45: At least the hindcast described in Weisse and Pluess is available at a higher resolution than mentioned here.

Action: We have made minor adjustments to the text, as Weisse and Plüß (2006) have a finer, yet unstructured resolution.

16. Lines 47-51: There are different parameters mixed-up (e.g., resolution of tidal and bathymetric data sets are compared). Please revise.

Answer: We agree, this should be clarified, as tidal characteristic and bathymetric data, in this data set, are in fact different formats at different resolution.

Action: We added "as polyarea shape file" before tidal characteristic values.

17. Figure 2: The abbreviations EEZ and AWZ are mixed up.

Action: We replaced "AWZ" with "EEZ" in Fig. 2 (Fig. 1 in revised manuscript).

18. Section 2.2: What is the vertical resolution of the hydrodynamic model? Please add. When currents are discussed, are depth-averaged currents used? Please be specific.

Answer: The vertical resolution of the model is given before Figure 3. As we have 54 z-layers with varying depths, we would rather not elaborate any further. All data-products are depth-averaged which is indeed missing in the manuscript.

Action: We have added depth-averaged at the beginning of Sect. 2., in Sect. 3.3 and in the abstract.

19. Line 182: I assume that the meteorological forcing is not communicated between these two models. Please revise. What are the wave effects on hydrodynamics accounted for in the coupling?

Answer: Technically speaking, the HN-model does transfer its meteorological information to the wave model, although we agree that this information is redundant here. The two-way coupling transfers wave radiation stress from UnK to UnTRIM². The other way, sea surface and current are considered in wave build up and dissipation as stated in the manuscript.

Action: We have removed communication of meteorological forcing from the manuscript and added which parameters are transferred between both models. Another sentence was added which stresses why a two-way coupled model was necessary for waves in the shallow areas of the German Bight at the end of Sect. 2.3.

20. Line 223: Salinity is shown, not salt transport.

Action: Changed "salt transport" to "salinity".

21. Line 290: Not the model validity is shown, only the extent to which model parameters considered agree with observations.

Action: We revised the first sentence of Section 4 to "In the following, we show the model's agreement with measurements for the years 1996 to 2015 using harmonic and tidal characteristic analysis".

22. Section 4.1: While M2 is the dominant constituent, shallow water effects are probably important in the model domain. A closer look at other components such as M4 would add value to the discussion.

Answer: We understand the desire for more tidal constituents, especially in the shallow water domain. Nevertheless, shallow water tides in the German Bight are numerous and their amplitudes rarely exceed 5 cm which would yield poor agreement to measurements. We would like to refer you to our annual short validations and the validation document BAW Technische Berichte (2020) which assesses many shallow water constituents in the year 2012. It can also be argued that the agreement of flood duration between observation and prediction already indicates that tidal asymmetry and shallow water constituents are reproduced following the argumentation of Friedrichs (1988) for e.g. M4.

23. Line 320 "This led to worse error metrics …" I don't understand the argument.

    Action: We removed the sentence, because Table 1 shows similar agreement throughout all years, including 2006.

24. Line 339: Shifts in tide-gauge locations should be documented.

    Action: The argument was removed from the manuscript.

25. Discussion of Figures 10 and 11: This implies that most of the bias is due to errors in tidal low waters. This should be mentioned explicitly.

    Answer: We disagree, because we describe the RMSE (see Eq. 1) which represents the root of averaged, squared differences, instead of bias. In the RMSE, all information concerning over- or underestimation is lost due to squared differences, which is why a RMSE only indicates that an error is present. For this reason, we performed a quantile analysis of water levels in BAW Technische Berichte (2020), as well as in our annual short validation documents. As we believe this a misunderstanding concerning bias and RMSE, we will not conduct any changes to the manuscript.

26. Figure 13: The slope needs a unit. Discussion of R-squared needs to be clarified (see major comment #6)

    Answer: The slope $m$ does not have a unit (in $y=mx+b$, $b$ and $x$ have a unit). The discussion concerning $R^2$ has hopefully been resolved in our answer to the major comment 6.

27. Table 2 and the discussion in the text: Is Tm02 or mean wave period discussed? Also, how useful are the error statistics for the peak period? Frequencies are discretized differently in both models and values can only take discrete numbers.

    Answer: In Table 2 and the discussion we refer to $T_{m02}$, which may be regarded as the mean wave period (see first sentence in Sect. 4.4). For completeness of the calculated spectral wave parameter we included the error statistics for the peak period $T_p$ as well. $T_p$ is a vital spectral wave parameter and hence should be included in the discussion. We do recognize though, that the peak wave period is misrepresented at calm-weather conditions, as neither UnK nor SWAN have open boundary wave forcing to the northern Atlantic which prohibits the propagation of North Atlantic swell into the model area. Our answers 1 and 2 to Review #1 also discuss this issue.

    Action: We added more information on the quality of error metrics for peak periods in calm weather conditions. For more detail see the answer to Reviewer #1.

28. Line 478 "relevant forcing parameters" Forcing of what?

Action: Relevance depends on the application which is why we have removed "relevant" from the manuscript. We have added some examples of forcing parameters to the revised manuscript (fresh water discharge, wind speed, and tidal dynamics).

29. Line 484 "It should be mentioned … it is hoped …". This is very vague. Is it possible to be more specific?

Answer: We have now added specific examples, which resulted from our stakeholder involvement. "The early involvement of potential stakeholders has shown potential uses apart from scientific applications. The range of applications covers coastal engineering projects such as the planning of offshore wind farms to support environmental tasks such as the description of habitats for the European Marine Strategy Framework Directive". See our answer to major point 7.

30. Line 491: Which part or applications of sea level science are meant?

Answer: We referred to the assessment of the effect of a rise in mean sea level on tidal dynamics, which will be overshadowed by other effects (e. g. differences in meteorological forcing, nodal tide modulation).

Action: changed sea level science to "studying the effect of a rise in mean sea level".

31. Line 491 "Thus, we propose …" What exactly means proposed? Is there an ongoing attempt? If so, please specify.

Action: We have clarified that a continuation is wishful thinking at the moment.

32. There are several typos or uncommon phrases. I suggest that the manuscript could benefit from some language editing.

Answer: As non-native English speakers we take your remark very serious and have carried out professional language editing which is why our response is delayed.

Action: Minor language adjustments all over the text.

---

## Author Response (AR2)

Dear Dr. Giuseppe M.R. Manzella,

thank you for the reviewing our revised manuscript and your remarks concerning the first part of our publication.

We have made the following minor changes to the manuscript:

1. Abstract, line 16: I strongly suggest to delete 'second'. The sentence will be: This part of two-part publication ...

   *changed in the manuscript*

2. line 40: Earth scieneces (capitol e)

   *changed in the manuscript*

3. line 66: reference Sievers et al (2021 ... delete parenthesis before 2021

   *changed in the manuscript*

4. line 68: delete 'which is the first part of this two part publication'

   *changed in the manuscript*

5. line 68: substitute 'In the first part ... ' with 'In Sievers et al. (2021) ..

   *changed in the manuscript*

We look forward to hearing from you soon.

Sincerely,

Robert Hagen on behalf of the authors